# Discovering Hierarchical Latent Capabilities of Language Models via Causal Representation Learning

## Abstract

Deriving actionable insights from language model evaluations to guide post-training is a central challenge, hampered by complex confounding effects and the prohibitive cost of controlled studies. In this paper, we propose a causal representation learning framework to uncover a *hierarchy* of LLM capabilities purely from publicly available observational data. Drawing insights from recent factor analysis (Ruan et al., 2024), we model the observed benchmark performance as a linear transformation of a few latent capability factors. Crucially, we model these latent factors as causally interrelated after appropriately controlling for the base model as a common confounder. Applying this approach to a comprehensive dataset encompassing over 1500 models evaluated across six benchmarks from the Open LLM Leaderboard, we identify a concise three-node linear causal structure that reliably explains the observed performance variations. The hierarchy that we discover — from general problem-solving to instruction-following, and finally to mathematical reasoning — is more than an interpretation; it provides a causal roadmap for post-training. We demonstrate through targeted fine-tuning experiments that interventions on parent capabilities propagate to child capabilities as predicted by our model, offering validated, actionable guidance for practitioners.

## 1 Introduction

State-of-the-art large language models (LMs) have exhibited exceptional proficiency across a wide spectrum of intricate natural language processing tasks, encompassing text generation, summarization, question answering, and creative language synthesis (Brown et al., 2020; Achiam et al., 2023; Grattafiori et al., 2024; Anthropic, 2024; Abdin et al., 2024; Yang et al., 2024; Guo et al., 2025). These billion-parameter models are often pre-trained extensively on diverse web corpora and undergoes various post-training stages including supervised fine-tuning (SFT), reinforcement learning with human feedback (RLHF) (Ouyang et al., 2022; Bai et al., 2022) to enable downstream model deployment. These complicated system engineerings make it hard to evaluate how models acquire capabilities and derive scientific claims thereafter.

In particular, rigorous understanding of post-training effects presents notable difficulties: (i) **Costs and heterogenity in pre-training**: implementation details such as data mixture, model architecture, etc, are often proprietary and vary greatly across institutions. For example, models might be subject to contamination on benchmark data (Golchin & Surdeanu, 2023); the heterogeneity of base models leads to evidence that the benefits of post-training on reasoning abilities can differ substantially even between models of comparable size (Gandhi et al., 2025; Zhao et al., 2025; Hochlehnert et al., 2025). Even with transparent pre-training recipes, training from scratch to control for these confounders through rigorous controlled studies implies prohibitive costs (Cottier et al., 2024; Qi et al., 2025). (ii) **Intricate interdependencies among distinct capabilities** — such as reasoning, few-shot learning, and instruction-following further complicates evaluation. For example, fine-tuning on instruction data might not improve knowledge-intensive question answering capabilities (Ghosh et al., 2024; Gudibande et al., 2023). Although various capabilities often seem to co-emerge as model scale increases (Ouyang et al., 2022; Zhang et al., 2025; Chen et al., 2025; Liu et al., 2025), systematic understanding of their *interactions* is still lacking.

In this paper, we introduce a novel approach to tackle these challenges by relying solely on publicly available evaluation data. We introduce a framework for modeling capability factors through an explicit structured representation. The cornerstone of our methodology rests on a crucial insight: the heterogeneity observed across diverse domains (Huang et al., 2020; Jin & Syrgkanis, 2024; Zhang et al., 2024b) — rather than being merely an obstacle — actually provides valuable "multi-view" perspectives into the shared latent capability structures across different base models. This lens enables us to identify and characterize these underlying structures with strong guarantees. Importantly, compared with existing factor modelling approaches (Ruan et al., 2024; Truong et al., 2025) that focus solely on the *interpretation* of evaluation data, **the structural information that we acquire directly provides insights into post-training practices**, as we demonstrate through expriments.

To establish the theoretical foundation for our investigation, we propose two hypotheses. We will show in Sections 2 and 3 that it is possible to quantify whether they indeed hold for our data.

1. **Capability-Performance Invariance:** A small, distinguishable set of latent capability factors governs benchmark performance, maintaining consistent relationships across diverse base models.

2. **Hierarchical Capability Structure:** Within any individual base model, these capabilities organize themselves into a hierarchical framework representable as a directed acyclic graph (DAG) (Pearl, 1995). In this structure, an edge $A \rightarrow B$ signifies that interventions targeting capability $A$ can propagate through the model's internal mechanisms to influence capability $B$, revealing causal pathways of skill development.

While the first hypothesis has been explored in a series of recent studies (Ruan et al., 2024; Ren et al., 2025; Polo et al., 2024), the second represents a novel contribution to the literature, although the hierarchical structure of *human* capabilities has been an active and influential research area in philosophy Simon (2012), cognitive science Carroll (1993); Anderson (1996); Anderson et al. (2004); Kaufman et al. (2009); Tenenbaum et al. (2011) and neuroscience Koechlin et al. (2003); Badre & D'Esposito (2007). We formalize a common intuition among practitioners: capabilities are not independent targets for fine-tuning. Improving a foundational skill like instruction-following might be a prerequisite for, and causally influence, a more specialized skill like mathematical reasoning.

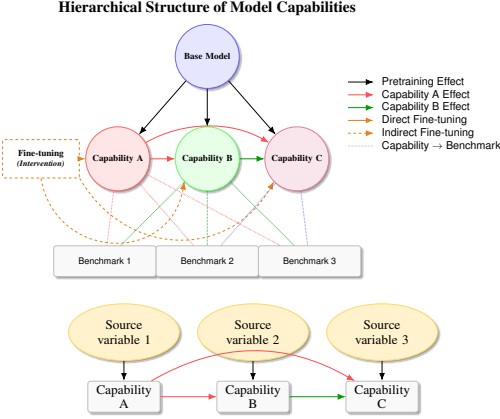

Figure 1: Example of a Hierarchical model of capabilities influencing benchmark performance (top) and hypothesized mechanism (bottom).

We formalize this hierarchical capability structure within Pearl's structural causal model framework Pearl (1995), treating the base model as a shared latent parent that influences all capability factors and fine-tuning as an intervention on these latent factors, as illustrated in Figure 1. Under this structural hypothesis, existing unstructured factorization approaches (such as PCA) for analyzing latent capabilities (Ruan et al., 2024; Ren et al., 2025; Polo et al., 2024) may fail to disentangle hierarchical latent factors due to their lack of causal constraints. Probabilistic latent-variable approaches, such as Item Response Theory (IRT) models (Truong et al., 2025) and Bayesian latent factor models (Papastamoulis & Ntzoufras, 2022), require full likelihood specifications and hand-crafted modeling assumptions. Moreover, all these approaches fail to account for the base model's overarching influence on all latent capabilities. Drawing inspiration from the causal representation learning (CRL) literature, we propose **Hierarchical Component Analysis (HCA)** that exploits heterogeneity across base models to recover hierarchical latent capabilities with provable identifiability guarantees under mild conditions.

We apply HCA to the open LLM leaderboard data[1] and show that models fine-tuned from four base models: Qwen2.5-7B, Qwen2.5-14B, Llama-3-8B, and Llama-3.1-8B can be well-explained by a linear SCM. We further assign meaningful semantic interpretations to these factors, allowing practi-

---

[1] https://huggingface.co/spaces/open-llm-leaderboard/open_llm_leaderboard#/

tioners a clear understanding of which capabilities to target during fine-tuning. Indeed, establishing explicit alignments between learned latent factors and human-interpretable concepts has remained both a significant and underexplored area within the CRL literature. To address this gap, we systematically explore correlations between latent factors, established benchmarks, and the effectiveness of prevalent leaderboard interventions. Moreover, performance on the general-reasoning parent node – encompassing benchmarks such as MMLU (Hendrycks et al., 2020) and BIG-Bench-Hard (Suzgun et al., 2022) – correlates more strongly with base-model FLOPs, underscoring the importance of scaling up pre-training compute for downstream problem solving.

## 1.1 NOTATION

Most analysis of this work is based on the open LLM leaderboard, which contains the accuracy of $N_0 = 4576$ LMs on $d = 6$ benchmarks. The leaderboard does not directly provide information on base model,[2] so we develop a principled approach to determine the base model from information in other columns. In this way, we get a subset of models on the original leaderboard that contains $N = 3360$ LMs with known base models, which we denote by $\Theta$. The eight most frequently-used base models includes Llama-3-8B, LLama-3.1-8B, (Grattafiori et al., 2024), Qwen2.5-0.5/7/14B Yang et al. (2024), Qwen2-7B (Yang et al., 2024), Mistral-7B (Jiang et al., 2023) and Gemma-2-9B (Team et al., 2024). Some parts of our analysis also include other base models into study, which we will explicitly describe. We will denote these eight base models by $\theta_1^*, \theta_2^*, \cdots, \theta_8^*$. We let $\Theta_k = \{\theta_{1,k}, \cdots, \theta_{N_k,k}\} \subseteq \Theta$ be the set of models using $\theta_k^*$ as base model.

For any LM $\theta$ and benchmark B, we use $x_{\theta,B}$ to denote the accuracy of $\theta$ on B, if observed. In our setting, we observe $x_{\theta_{i,k},B_j}$ for all $k \in [K], i \in [N_k]$ and $j \in [d]$, and we will simply denote this by $x_j^{(k,i)}$. Then $\boldsymbol{x}^{(k,i)} = \left(x_j^{(k,i)}\right)_{j=1}^d$ is the observed performance vector for model $\theta_{i,k}$. The set of all data, $\left\{\boldsymbol{x}^{(k,i)} : k \in [K], i \in [N_k]\right\}$, is denoted by $\boldsymbol{X}$. We also define $\boldsymbol{X}^{(k)} = \left\{\boldsymbol{x}^{(k,i)} : i \in [N_k]\right\}$. In what follows, we will sometimes abuse notation and view $\boldsymbol{X}' \subseteq \boldsymbol{X}$ as a $|\boldsymbol{X}'| \times d$ matrix, where each row is a performance vector $\boldsymbol{x}_j \in \boldsymbol{X}'$.

## 2 THE LATENT CAPABILITY HYPOTHESIS

Recently, a line of works developed observational scaling laws (Ruan et al., 2024; Ren et al., 2025; Polo et al., 2024). The key hypothesis that make their analyses possible is that the observed benchmark performance is some linear transformation of low-dimensional latent capability vectors.

**Hypothesis 0.** *There exists some latent capability vector $\boldsymbol{z} \in \mathbb{R}^{d_0}, d_0 < d$ and some matrix $\boldsymbol{G} \in \mathbb{R}^{d \times d_0}$, such that $\boldsymbol{x} = \boldsymbol{G}\boldsymbol{z}$.*

Principal component analysis (PCA) provides a systematic approach to validate this hypothesis by examining whether the model-accuracy matrix exhibits an approximate low-rank structure. Applying PCA to the leaderboard data, we find that the performance matrix is approximately rank-3 (as shown in Figure 2a), aligning closely with previous findings reported by existing works.

We next examine whether these patterns persist when we vary the model subset used for PCA. To this end, we isolate all Qwen2.5 variants (across scales) and, separately, the Qwen2.5-14B model. As shown in Figure 2b, the distributions of their distances from the full-leaderboard rank-3 PC subspace diverge substantially. This divergence implies that a one-size-fits-all PCA—applied indiscriminately to every model—can obscure meaningful heterogeneity unique to particular model families.

To further examine this heterogeneity, we choose the eight most commonly used base models on the leaderboard as listed in Section 1.1, and examine the PC subspaces of their corresponding domains. Specifically, for each $k = 1, 2, \cdots, 8$, we apply PCA to the domain data $\boldsymbol{X}_k$ to obtain the rank-3 principal component subspace, and then measure the cosine distances between these subspaces. The similarity matrix is shown in Figure 2c, revealing a striking pattern: five domains (with base models Llama-3-8B, Llama-3.1-8B, Qwen2-7B, Qwen2.5-7/14B) have roughly the same PC subspaces, whereas the other three lie distinctly apart. We define $S_{\text{inv}} = \{1, 2, 4, 5, 6\}$ to be the index set of these seven models and $\boldsymbol{X}_{\text{inv}} = \cup_{k \in S_{\text{inv}}} \boldsymbol{X}_k$ to be the corresponding benchmark performance data. Notably, this heterogeneity persists under ICA (Hyvärinen et al., 2009), another popular factor

---

[2]There is a column named 'Base model' in the leaderboard, but usually that base model is itself a fine-tuned version of, say, Llama-3-8B.

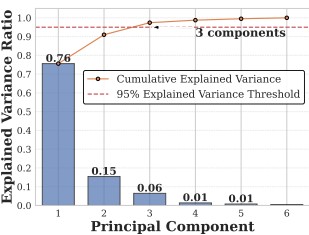
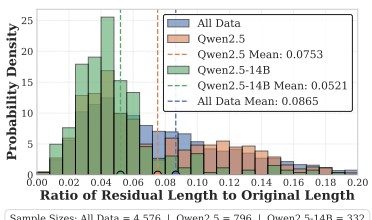
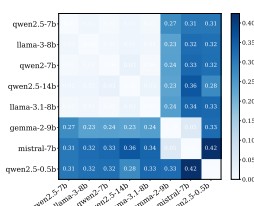

(a) Low-rank structure of the leaderboard data.

(b) Distance distributions to the principal component subspace.

(c) Principal component subspace similarity across domains.

Figure 2: PCA analysis showing low-rank structure, domain heterogeneity, and subspace similarity in leaderboard data.

analysis method, since PCA and ICA span the same component subspace, differing only in how they parametrize the independent sources within it.

Overall, we have found that:

- Performance data exhibit distinct heterogeneity among models fine-tuned from different base models. This underscores the necessity of controlling for base models when interpreting statistical results from benchmark evaluations.

- A particular subset of base models emerges, wherein their fine-tuned derivatives consistently reveal similar latent performance patterns across benchmarks.

In what follows, we build on these observations and introduce a novel latent factor model for LM capabilities.

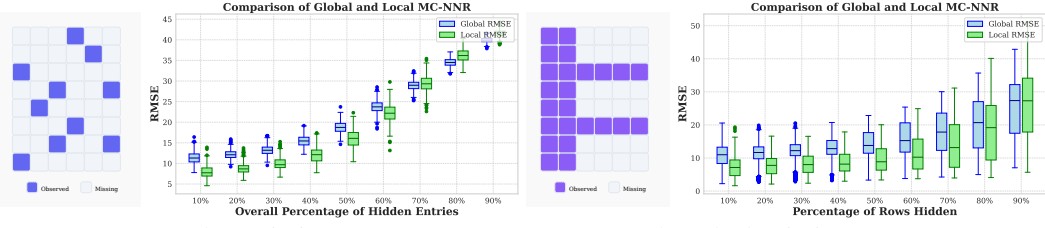

(a). Random missing pattern.

(b). Block missing pattern.

Figure 3: RMSE of global and local matrix completion approaches for two types of missing patterns.

**Remark 1.** *As a brief detour, we show how our findings can help us more accurately impute missing benchmark performance in the leaderboard. We focus on two distinct missing patterns, illustrated in Figure 3, and we are interested in the missing performances of models fine-tuned on Qwen2.5-14B. Motivated by the heterogeneity we observed in the PC subspaces, we apply matrix completion with nuclear norm regularization (MC-NNR) both within the Qwen2.5-14B domain and across the entire leaderboard. We find that the former approach yields notably lower reconstruction error. Additional details are provided in Figure 15.*

## 2.1 A REFINED HYPOTHESIS

In view of the limitation of Hypothesis 0, we propose the following modification, restricting it to domains with identical PC subspaces the we identify in Figure 2c:

**Hypothesis 1.** *The observed benchmark performance $\boldsymbol{x}_i \in \boldsymbol{X}_{\text{inv}}$ is governed by a set of latent capability factors $\boldsymbol{z}_i \in \mathbb{R}^{d_0}$, where $d_0 \leq d$. Moreover, there exists a linear and injective relationship between $\boldsymbol{z}_i$ and $\boldsymbol{x}_i$, meaning that there exists some matrix $\boldsymbol{G} \in \mathbb{R}^{d \times d_0}$ with full column rank such that $\boldsymbol{x}_i \approx \boldsymbol{G}\boldsymbol{z}_i, \forall i \in \boldsymbol{X}_{\text{inv}}$.*

In the remainder of this work, we will focus on the base models in $S_{\text{inv}}$ and their corresponding benchmark performance data $\boldsymbol{X}_{\text{inv}}$.

# 3 LEARNING HIERARCHICAL LANGUAGE MODEL CAPABILITIES

In this section, built upon the initial observations in Section 2, we introduce a new approach to capture invariant laws underlying different domains. Our approach leverages a latent hierarchical structure among different capability components in $z_i$. To formally describe this latent structure, we introduce the following definition of linear structural causal models (SCMs) (Pearl, 1995).

**Definition 1.** *Given a directed acyclic graph (DAG) $\mathcal{G} = (\mathcal{V}, \mathcal{E})$ with node set $\mathcal{V} = [d_0]$ and edge set $\mathcal{E}$, a linear SCM is a data-generating process of $d_0$ random variables $z_1, z_2, \cdots, z_{d_0}$ with $z_i = \sum_{j \in \mathrm{pa}_{\mathcal{G}}(i)} w_{ji} z_j + \sigma_i^{1/2} \epsilon_i, i \in [d]$ with independent source variables $\epsilon_i$ wuth unit variance, where $w_{ij} \in \mathbb{R}$ are weights and $\mathrm{pa}_{\mathcal{G}}(i)$ is the parent set of $i$ in $\mathcal{G}$.*

Intuitively, latent factors earlier in the topological ordering of the DAG are primitive, while later factors are progressively less primitive, as they inherits the variability in their ancestors.

In practice, assuming exact SCMs is often too restrictive. We define inexact SCMs below, which allows the source variables to be entangled with each other:

**Definition 2.** *A linear $\alpha$-inexact SCM is a data generating process of $z_1, z_2, \cdots, z_{d_0}$ with $\hat{\boldsymbol{\epsilon}} = \boldsymbol{U}\boldsymbol{\epsilon}, z_i = \sum_{j \in \mathrm{pa}_{\mathcal{G}}(i)} w_{ji} z_j + \sigma_i^{1/2} \hat{\epsilon}_i, i \in [d_0]$ for some independent source variables $\epsilon_i$ wuth unit variance and some matrix $\boldsymbol{U} = [\boldsymbol{u}_1, \cdots, \boldsymbol{u}_{d_0}]^\top \in \mathbb{R}^{d_0 \times d_0}$ with $\|\boldsymbol{u}_i\|_2 = 1$ and $\frac{1}{d_0} \sum_{i \neq j}^{d_0} u_{ij}^2 \leq \alpha$. Finally, for a collection $\mathcal{C}$ of $\alpha_i$-inexact linear SCMs sharing the same causal graph $\mathcal{G}$, we define $\alpha = \max_i \alpha_i$ to be the maximum inexactness coefficient (MIC) of $\mathcal{C}$.*

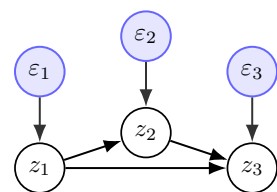

(a) A standard SCM with $z_i$'s being the causal factors.

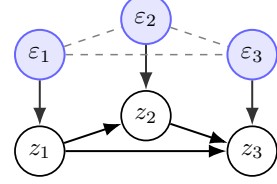

(b) An inexact SCM where the $\epsilon_i$'s can be dependent.

Figure 4: Illustration of Definition 1.

When $\alpha = 0$, an $\alpha$-inexact SCM becomes an exact SCM. Hence, the MIC measures the extent of violating the independence assumption on the source variables. Given this definition, we are ready to state our second hypothesis. A graphical illustration of exact and inexact SCMs is given in Figure 4.

**Hypothesis 2.** *There exists a subset of domain indices in $S \subseteq S_{\mathrm{inv}}$, such that for all $k \in S$, the capability factors $z^{(k,i)}$ associated with $x^{(k,i)} \in X_k$ are generated from linear inexact SCMs with some small MIC, and the causal graph $\mathcal{G}$ is invariant across all $k$'s, while the weights $w_{ij}^{(k)}$ and errors $\hat{\epsilon}_i^{(k)}$ can be domain specific.*

Different from all existing works that are restricted to correlation-based analysis, Hypothesis 2 characterizes a *causal* generative mechanism underlying an LM's capabilities. Specifically, given a base model $\mathsf{B}_k$, each independent factor $\epsilon_i$ directly influences exactly one capability $z_i$, while other capabilities are either unaffected by $\epsilon_i$ or affected only *indirectly* through $z_i$.

Since Hypothesis 1 and Hypothesis 2 need not hold for every data distribution, it is necessary to develop a diagnostic that empirically tests their validity in our context. Once this is done, we pursue two objectives: (1) recover the latent capability factors that drive observed benchmark performance, and (2) characterize precisely how those capabilities map to performance outcomes.

It turns out that these questions are closely related to recent advances in multi-domain causal representation learning (CRL) Jin & Syrgkanis (2024); Zhang et al. (2024b). In that setting, one assumes $K$ domains $\mathfrak{E} = \{\mathcal{E}_k : k \in [K]\}$ and a dataset $\boldsymbol{X}^{(k)} = \{\boldsymbol{x}^{(k,i)}\}_{i=1}^{N_k}$ associated with the $k$-th domain generated from the structural equations

$$\boldsymbol{z}^{(k,i)} = \boldsymbol{A}_k \boldsymbol{z}^{(k,i)} + \boldsymbol{\Omega}_k^{1/2} \boldsymbol{\epsilon}^{(k,i)}, \quad \boldsymbol{x}^{(k,i)} = \boldsymbol{G}\boldsymbol{z}^{(k,i)}, \qquad k \in [K], \tag{1}$$

where $(\boldsymbol{A}_k)_{ij} = w_{ij}^{(k)}$ if there exists a direct causal edge $z_j \to z_i$ in the latent graph $\mathcal{G}$ and otherwise it is zero, $\boldsymbol{\Omega}_k$ is a diagonal matrix encoding the variances of source variables. $\boldsymbol{G}$ is the shared mixing matrix. For convenience, we assume that the nodes of $\mathcal{G}$ is sorted in a topological order, *i.e.*, $z_j \to z_i$ implies $j < i$. CRL seeks to uncover both the causal graph $\mathcal{G}$ and the mixing map $\boldsymbol{G}$. Table 1 summarizes the parallels and distinctions between this CRL framework and our LM capability model.

| Causal representation learning | Our context: learning latent LM capabilities |
|---|---|
| Domain set $\mathfrak{E}$ | Each domain $E_k \in \mathfrak{E}$ is defined by a base model $\mathcal{M}_k$ and the observed dataset $\boldsymbol{X}_k$ that contains the performance of all LM $\{\theta_{i,k}\}_{i=1}^{N_k}$ that use $\theta_k^*$ as base model. |
| Observed $\boldsymbol{X}^{(k)} = \{\boldsymbol{x}^{(k,i)}\}_{i=1}^{N_k}, k \in [K]$ | $\boldsymbol{x}^{(k,i)} \in \mathbb{R}^d$ contains the known benchmark accuracies of $\theta_{i,k}$. |
| Causal factors $\boldsymbol{Z}^{(k)} = \{\boldsymbol{z}^{(k,i)}\}_{i=1}^{N_k}, k \in [K]$ | $\boldsymbol{z}^{(k,i)}$ is the unobserved $d_0$-dimensional capability vector of $\theta_{i,k}$ that possesses some causal structure. We assume that $d_0 \leq d$. |
| Mixing matrix $\boldsymbol{G}$ (invariant across different domains) | The observed benchmark performance is a linear transformation of the underlying capability factors. This linear dependency does not change no matter what base model is chosen. |
| Identification of exact causal models | We define the notion of inexact causal models, and the objective is to minimize the inexactness. |

Table 1: A comparison between linear CRL and some key elements in our context.

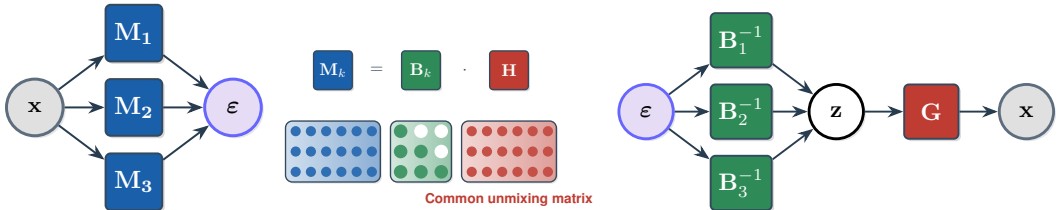

(a) ICA mapping $\mathbf{x}$ to source variables $\boldsymbol{\varepsilon}$.
(b) Decomposition of $\mathbf{M}_k$ that we need to recover.
(c) The whole data generating process, where $\boldsymbol{G} = \boldsymbol{H}^\top(\boldsymbol{H}\boldsymbol{H}^\top)^{-1}$ is the mixing matrix.

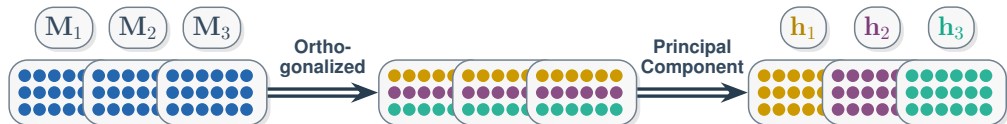

(d) Finding orthogonalized matrices and extracting principal components.

Figure 5: Illustration of our setting and the key row-residual extraction step in our algorithm

Prior work of Jin & Syrgkanis (2024) showed that for exact linear SCMs, assuming that the domains $\mathcal{E}_k$ satisfy a richness assumption, the latent causal factors are identifiable up to a benign ambiguity set, which for instance implies that one can recover the mixing matrix $\boldsymbol{G}$ up to a left multiplication of lower-triangular matrix for the causal model in Figure 4. However, the identification algorithm presented in Jin & Syrgkanis (2024) is sensitive to the exactness assumption. We propose a novel identification algorithm, which we call Hierarchical Component Analysis (HCA), that is more robust to the inexactness of the SCM. The main ideas of HCA are discussed in Section 3.1, and more details can be found in Appendix C.

## 3.1 HIERARCHICAL COMPONENT ANALYSIS (HCA)

To move beyond simple correlation and recover the causal hierarchy outlined in Hypothesis 2, standard factor analysis is insufficient. We now introduce the main ideas behind HCA, an algorithm designed to leverage heterogeneity across base models to identify the underlying directed structure.

**1. ICA-based unmixing.** As a first step, we apply Independent Component Analysis (ICA, (Hyvärinen et al., 2009)) separately to each domain $k \in [K]$ to obtain an unmixing matrix $\boldsymbol{M}_k$ that maps independent source variables to observed benchmark data $\boldsymbol{x}$, as shown in Figure 5a. Under standard non-Gaussian assumptions in the ICA literature, these source varisbles are uniquely identified as $\boldsymbol{\epsilon}^{(k)}$ up to permutations, implying that $\boldsymbol{M}_k = \boldsymbol{P}_k \boldsymbol{B}_k \boldsymbol{H}$, where $\boldsymbol{P}_k$ is an unknown permutation matrix, $\boldsymbol{B}_k = \boldsymbol{\Omega}_k^{-1/2}(\boldsymbol{I} - \boldsymbol{A}_k)$ is lower-triangular, and $\boldsymbol{H} = (\boldsymbol{G}^\top\boldsymbol{G})^{-1}\boldsymbol{G}^\top$ is the right inverse of $\boldsymbol{G}$. Our goal is to recover the matrices $\boldsymbol{B}_k$ and $\boldsymbol{H}$ from $\boldsymbol{M}_k$, as they allow us to recover the whole DGP as shown in Figure 5c. In partcular, the latent factors are recovered via $\boldsymbol{z} = \boldsymbol{H}\boldsymbol{x}$.

**2. Row-residual extraction.** For any matrices $M_k, k \in [K]$, we derive a testable equivalent condition for admitting the decomposition $M_k = B_k H$. specifically, for each component index $i \in [d_0]$, we can compute the residual $r_{k,i}$ of projecting the $i$-th row of $M_k^*$ onto the span of its first $(i-1)$ rows. Then such decomposition exists if and only if $[r_{k,i}]_{k=1}^K$ is rank 1 for all $i$, and $h_i$ can be recovered (up to scale) as its principal singular vector. This process is visualized in Figure 5d.

**3. Permutation alignment and factor refinement.** Since each $M_k^*$ is known only up to row permutation, we search over all permutations of the rows of $M_k$. For each case, we apply the previous step to obtain an estimate of $H$, and then refine each domain's weight matrix by solving $\min_{B_k \text{ lower-triangular}} \|M_k - B_k H\|_F^2$, thereby fitting the best hierarchical structure to the observed unmixing matrices. Finally, we choose the set of permutations that induces minimal MIC.

The full description HCA appears in Algorithm 2, and Appendix C.2 proves that, under an exact SCM, HCA is guaranteed to identify the underlying causal factors up to some benign ambiguities. Specifically, for the causal graph in Figure 4, $H$ is recovered up to a left multiplication of lower-triangular matrix. Equivalently, each identified latent causal factor for $z_i$ is a mixture of $z_j, 1 \leq j \leq i$. As shown in Jin & Syrgkanis (2024), this ambiguity is not a limitation but rather an intrinsic property reflecting equivalent models that generate identical distributional outcomes.

When the SCM is inexact, HCA recovers a data generating process

$$z^{(k,i)} = \hat{B}_k^{-1} \hat{\epsilon}^{(k,i)}, \quad x^{(k,i)} = \hat{G} z^{(k,i)}, \quad k \in [K],$$

so that $\hat{\epsilon}^{(k,i)} = \hat{B}_k \hat{H} x^{(k,i)}$. On the other hand, the ICA recovers $\epsilon^{(k,i)} = M_k x^{(k,i)}$ with independent source components, so one can see that $\hat{\epsilon}^{(k,i)} = J_k \epsilon^{(k,i)}$ where $J_k = \hat{B}_k \hat{H} M_k^\top (M_k M_k^\top)^{-1}$. This provides a guarantee on the MIC (introduced in in Definition 2):

**Proposition 1.** *Suppose that the ICA step is exact, then HCA recovers a linear $\alpha_k$-inexact SCM for the $k$-th domain, where $\alpha_k = \frac{1}{d_0} \sum_{i \neq j} (\tilde{J}_k)_{ij}^2$, $(\tilde{J}_k)_i = (J_k)_i / \|(J_k)_i\|_2$. It follows that $\alpha = \max_{k \in [K]} \alpha_k$ is a valid MIC.*

Proposition 1 provides a quantitative measure of how well the recovered causal model can explain the variations in the observed benchmark data $X^{(k)}, k \in [K]$.

## 4 EXPERIMENTAL RESULTS

### 4.1 RECOVERY OF THE DATA GENERATING PROCESS

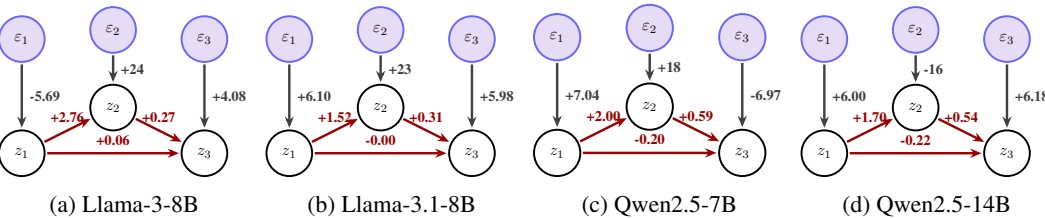

|  |  |  |  |
|---|---|---|---|
| (a) Llama-3-8B | (b) Llama-3.1-8B | (c) Qwen2.5-7B | (d) Qwen2.5-14B |

Figure 6: The causal graphs that we recover for each domain. The numbers represent the weights of each causal edge. For instance, in the Llama-3-8B domain, $z_2 = 2.76 z_1 + 24\epsilon_2$.

Given its theoretical justification in the previous section, we now use HCA to recover a causal model with $d_0 = 3$ nodes that explains the observed benchmark performance of models within domains in $S_{\text{inv}}$. We observe that running our algorithm on the subset of $\{1, 2, 4, 5\}$, with Qwen2-7B excluded, achieves a minimal MIC of $0.04$. This likely indicates that Qwen2-7B may deviate from the shared causal pattern of the other four base models (Llama-3-8B, Llama-3.1-8B, Qwen2.5-7B, Qwen2.5-14B). Moreover, in view of the ambiguity discussed in Section 3.1, we run an OLS $z_i \approx \sum_{j<i} a_j z_j + \gamma_B x_B + c$ where $x_B$ represents the performance on benchmark B. For each $i$, we pick B that maximizes the $R^2$ and replace $z_i$ with $z_i - \sum_{j<i} a_j z_j$ to attain best-possible alignment between the recovered latent factors and their most indicative benchmarks.

The causal graphs that we recover are shown in Figure 6. The source factors $\epsilon_i$'s are normalized to have unit variance. In Figure 7a, we present the unmixing matrix (*i.e.*, linear mapping from

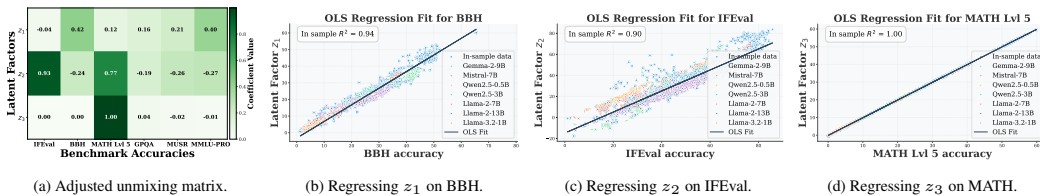

(a) Adjusted unmixing matrix.    (b) Regressing $z_1$ on BBH.    (c) Regressing $z_2$ on IFEval.    (d) Regressing $z_3$ on MATH.

Figure 7: The unmixing matrix and the alignment between benchmarks and capabilities via OLS. We also compare the fitted OLS with the latent factor values of other base models.

benchmarks to latent capabilities), from which interesting patterns can be observed: $z_1$ is a mixture of all five benchmarks except IFEval with BBH and MMLU-Pro contributing the most, $z_2$ is a mixture of IFEval and MATH Lvl 5, and $z_3$ is almost identical to MATH Lvl 5. We will revisit these observations in the next subsection. Figure 7 further shows the results of OLS, where $z_1, z_2, z_3$ are observed to correlate strongly with BBH, IFEval and MATH Lvl 5, respectively. It is also important to notice that the causal conclusions we draw only apply to the four base models being considered: Figure 7 shows that the fitted OLS can have poor performance on some other base models.

## 4.2 VALIDATING THE CAUSAL HIERARCHY AND DERIVING POST-TRAINING INSIGHTS

In the previous subsection, we explored the correlation between benchmark performance and the inferred latent factors. However, practically interpreting what a causal intervention entails within this framework remains unclear. The broader challenge of interpreting and intervening on latent factors is a longstanding and unresolved issue within causal representation learning, with no universal methodology currently available. By further analyzing the Llama-3 and Qwen2.5 models included in leaderboards, we propose hypotheses regarding these latent capabilities, supported by reliable empirical evidence.

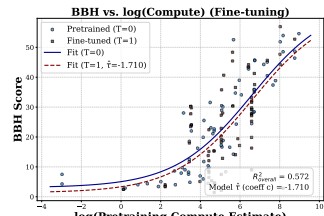

Figure 8: Sigmoid scaling law for BBH performance.

**Interpreting $z_1$ (Foundational General Capability).** As a root node in the causal graph, $z_1$ a root node in our causal graph, likely represents a foundational, generalized capability. This interpretation is supported by its positive influence across nearly all benchmarks (see mixing matrix in Figure 7a), consistent with the expectation that enhancing a general capability should broadly improve downstream task performance. Interestingly, we find that model performances on benchmarks well-aligned with general capabilities, such as BBH, roughly follows a sigmoid scaling law described by: $Y \approx L/(1 + \exp(-k(\log C - \log C_0)) + \tau T + b$, where $L, k, C_0, b, \tau$ are unknown parameters, $C$ is the pretraining compute and $T$ is a binary variable distinguishing fine-tuned models ($T = 1$) from from pretrained-only models ($T = 0$) as shown in Section 4.2. This relationship suggests that LMs' general capabilities are predominantly determined by pretraining compute resources and experience comparatively modest enhancements during subsequent post-training procedures.

**Interpreting $z_2$ (Instruction Following).** $z_2$ strongly correlates with IFEval, suggesting it embodies instruction-following capability. We extract instruction-tuned models from the Open LLM leaderboard, excluding those specific for math reasoning, which act as proxies of intervention on $z_2$. These models show minimal changes on BBH, MMLU-Pro, GPQA and MUSR for the first three base models, aligning with its mixing pattern shown in Figure 7a. We also conduct supervised fine-tuning (SFT) directly on IFEval and observe similar patterns.

**Interpreting $z_3$ (Advanced Mathematical Reasoning).** $z_3$ highly correlates with the MATH Lvl 5 benchmark, suggesting it represents advanced mathematical reasoning capability. Isolating this capability through fine-tuning is challenging; targeted mathematical fine-tuning often causes catastrophic forgetting (Zhai et al., 2023), reducing performance on other tasks and likely affecting $z_1$ and $z_2$. Identifying fine-tuning strategies that selectively enhance mathematical reasoning ($z_3$) without negatively impacting other core capabilities ($z_1, z_2$) remains a critical open question. Furthermore, both observational instruct model performance and interventional SFT demonstrate substantial improvement on MATH, supporting the hypothesized causal link from $z_2$ (instruction-following) to $z_3$ (mathematical reasoning). This insight is highly practical: it suggests that to improve mathematical reasoning, one effective strategy is to first improve instruction-following capabilities. **This is a**

**non-obvious insight that our causal framework makes explicit.** Additionally, the causal effect of $z_2$ on $z_3$ is larger for Qwen models than for Llama models, consistent with the weights of the causal graphs shown in Figure 6.

## 5 DISCUSSIONS

In this work, we introduced a novel approach to discover the causal hierarchy of LLM capabilities purely from observational benchmark data. We moved beyond descriptive factor analysis by adopting insights from the causal representation learning literature. We conclude this work with a few takeaways and remarks that may be useful to practitioners.

| Model | Config | BBH | IFEval | MATH | GPQA | MUSR | MMLU-PRO |
|-------|--------|-----|--------|------|------|------|----------|
| Llama-3-8B | Base | 0.46 | 0.12 | 0.05 | 0.33 | 0.37 | 0.33 |
| | IFEval SFT | 0.49 | 0.50 | 0.05 | 0.32 | 0.38 | 0.33 |
| | Instruct | 0.51 | 0.53 | 0.11 | 0.30 | 0.40 | 0.35 |
| Qwen2.5-7B | Base | 0.54 | 0.33 | 0.23 | 0.32 | 0.44 | 0.44 |
| | IFEval SFT | 0.55 | 0.50 | 0.28 | 0.33 | 0.43 | 0.44 |
| | Instruct | 0.52 | 0.61 | 0.33 | 0.30 | 0.42 | 0.43 |
| Qwen2.5-14B | Base | 0.61 | 0.36 | 0.29 | 0.40 | 0.45 | 0.53 |
| | IFEval SFT | 0.63 | 0.55 | 0.32 | 0.36 | 0.43 | 0.52 |
| | Instruct | 0.64 | 0.82 | 0.55 | 0.32 | 0.41 | 0.49 |
| Gemma-2-9B | Base | 0.53 | 0.16 | 0.13 | 0.36 | 0.44 | 0.41 |
| | IFEval SFT | 0.52 | 0.63 | 0.12 | 0.33 | 0.42 | 0.40 |
| | Instruct | 0.59 | 0.57 | 0.17 | 0.34 | 0.42 | 0.41 |

Table 2: Performance comparison of language models before and after fine-tuning with IFEval SFT. BASE and INSTRUCT model results are sourced from the open LM leaderboard. Reported values represent averaged performance across all INSTRUCT models, excluding specialized math reasoning variants.

**For post-training evaluation:** Our results demonstrate that the impact of any fine-tuning intervention can differ substantially across base models. Evaluation studies therefore are expected to specify exactly which pre-trained checkpoints their methodology applies to. To quantify these heterogeneous effects, one can employ standard causal-inference tools – such as estimating the conditional average treatment effect (CATE) – to measure, with statistical rigor, how fine-tuning impacts performance on each base models.

**For model developers:** The directed, hierarchical structure of capabilities we uncover suggests a clear development priority: given sufficient compute budget, one can focus on scaling up pre-training FLOPs which are more correlated with upstream parent node $z_1$ performance and can, and gains there cascade to more specialized abilities. That said, not every capability is equally malleable. Some – like instruction-following – correlate less with model scale (i.e., FLOPs) and exhibit large noise-factor variances (the $z_2$ node in Figure 6), indicating they respond more readily to post-training. On the other hand, given limited budgets or for small models, our noise-weight estimates suggest that we may need other interventions like instruction tuning to further improve downstream performance.

**For model evaluators:** Due to the inherent hierarchical structure of evaluation suites, it is important to examine fine-grained performance beyond aggregate numerical scores. For example, gains on the MATH benchmark may partly stem from improved instruction-following, which, while related to, is not equivalent to the mathematical reasoning the benchmark aims to evaluate. Secondly, as specialized abilities are causally affected by upstream ones, evaluators can therefore prioritize designing benchmarks that evaluate general, foundational capabilities, such as BBH and MMLU-Pro. They reflect more substantive improvements rather than artifacts of limited domain adaptation.

**Limitations and future work:** Our analysis identified a three-factor hierarchy, a result constrained by the six benchmarks available in the Open LLM Leaderboard. **Our HCA method is not inherently limited to three capabilities; it can directly be used to discover a richer and more fine-grained hierarchy as more diverse evaluation data becomes available.** For instance, applying HCA to a suite of 20 diverse benchmarks could potentially reveal a more complex *skill graph*. This represents a promising direction for future work.

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

# Appendices

# Table of Contents

LLM USAGE

We use LLMs to aid and polish writing.

## A RELATED WORK

**Benchmark-driven LM capability studies.** Benchmarks give researchers a shared scoreboard, letting everyone check claims about better language models instead of relying on hype. Early scaling-law studies showed that test loss falls in a smooth power curve as model size, data, and compute grow, setting a baseline for how capability should rise (Kaplan et al., 2020). Later work found many frontier models were under-trained for their size and mapped out a compute-efficient path that the Chinchilla model follows (Hoffmann et al., 2022). Instead of running new model sweeps, Ruan et al. (2024) proposed observational scaling laws involving *latent capability factors*, that depend on the model family and are obtained by PCA. They showed that benchmark performances are inherently low-rank and 3 principal components are sufficient to obtain good fitting performance. This approach is also adopted by some follow-up works on new tasks (Ren et al., 2025) and larger sets of models (Polo et al., 2024). Dominguez-Olmedo et al. (2025) further proposed an adjustment of the scaling law based on the model release time, given the fact that later models are more likely to be "trained on test tasks".

While pretrained LMs exhibit predictable scaling laws post-training presents a more complex picture regarding such predictive capabilities. For fine-tuning, performance generally scales with model size and fine-tuning data (as suggested by Zhang et al. (2024a)), but the "transfer gap" between pre-training and downstream tasks is a key variable (Barnett, 2024), and pre-training metrics aren't always reliable predictors of post-tuning success. Instruction tuning demonstrates clear benefits from scaling model size and the number/diversity of instructional tasks, as shown by work on FLAN (Wei et al., 2022), T0 (Sanh et al., 2022), and FLAN-PaLM Chung et al. (2024). RLHF, crucial for aligning models with human preferences (Ouyang et al., 2022), shows performance gains with larger models and more feedback. However, recent work (Hou et al., 2024) indicates RLHF might scale less efficiently than pre-training, with potential diminishing returns from increased data or reward model size under fixed conditions.

Recently, evaluation of post-training has been shown to be unreliable (Burnell et al., 2023; Hochlehnert et al., 2025). While the Open LLM Leaderboard provides truthful evaluation of models across different benchmarks, the contamination issue may still prevent us to obtain a reliable assessment of model capabilities.

**Connections between LM capabilities.** Research increasingly shows that LM capabilities are not isolated but form a complex, interconnected system. Studies reveal strong synergies, such as the bidirectional enhancement between coding and reasoning abilities (Zhang et al., 2025; Bubeck et al., 2023), and how strong reasoning underpins mathematical problem-solving (Lewkowycz et al., 2022). Complex skills often arise from compositionality, where LMs combine simpler, foundational skills in novel ways (Yu et al., 2023; Arora & Goyal, 2023; Chen et al., 2023).

Evidence also points towards latent abilities or general factors influencing performance across diverse tasks (Liang et al., 2023; Polo et al., 2024). The nature of emergent abilities – skills appearing in larger models – is debated, with some questioning if they are genuinely novel or byproducts of other mechanisms (Wei et al., 2022; Schaeffer et al., 2023).

Also, there are significant trade-offs: efforts to enhance safety can sometimes reduce raw capability (Chen et al., 2025), and fine-tuning for one skill can lead to catastrophic forgetting of others (Zhai et al., 2023). Phenomena like inverse scaling further highlight these complex interactions (McKenzie et al., 2023). Finally, successful task transfer and in-context learning demonstrate that LMs leverage shared underlying mechanisms and representations across different tasks (Min et al., 2022; Brown et al., 2020), underscoring the deep interrelations among their varied skills.

**Causal representation learning.** Causal representation learning (CRL) aims to recover latent variables and mechanisms that remain stable under interventions and distribution shifts, thereby enabling robust prediction, reasoning, and control. Foundational position papers argue that learning disentangled *causal* factors is essential for machine intelligence rather than merely desirable for interpretability (Schölkopf et al., 2021; Wang & Jordan, 2022). Most existing works are devoted

to establishing identifiability of causal representations in realistic scenarios. Weakly supervised disentanglement shows that paired samples before/after unknown interventions are sufficient to identify factors without compromising downstream utility (Locatello et al., 2020; Brehmer et al., 2022). von Kügelgen et al. (2023) showed that a pair of single-node *hard* interventions on each latent factor is sufficient for full identifiability of the latent causal factors. Subsequent works generalize this to the case of single-node soft interventions (Zhang et al., 2023; Squires et al., 2023; Buchholz et al., 2023). Recently, there has been a surge of interest in studying identifiability under multi-node interventions, which is much more practical Jin & Syrgkanis (2024); Zhang et al. (2024b). Closely related to CRL, invariant Risk Minimization (IRM) and its game-theoretic variants formalize how multiple training environments can pin down causal predictors (Arjovsky et al., 2019; Ahuja et al., 2021).

## B  IMPLEMENTATION DETAILS

**Supervised Fine-tuning**. We use lm-eval-hardness to evaluate models before and after fine-tuning. We first test base model performance and observe that it can match the performance in Open LM Leaderboard. We train all models with standard hyper-parameters for SFT - 3 epochs, learning rate 2e-5. Moreover, noticing that the IFEval dataset lacks ground truth responses followed by the instructions, we query GPT-4 to generate responses with the prompt *"You are a helpful assistant evaluating instruction-following ability. For each prompt, provide ONLY a direct response to the specific instruction, prefixed with 'Response: '. Keep your response concise, clear, and strictly follow the instruction without adding explanations or unnecessary information. Your response (excluding the 'Response: ' prefix) should strictly satisfy the length requirement."* Moreover, we also SFT on $z_1$ BBH. But we observe a marginal improvement over the same BBH test sets. We hypothesize that parent nodes like $z_1$ are more dependent on base model FLOPs thus maybe hard to improve through fine-tuning alone.

**Matching models on the leaderboard with the base models**. Our algorithm for mapping LLMs to their pretraining token counts implements a hierarchical, multi-layered identification strategy with progressively decreasing confidence levels. The approach consists of four distinct identification layers:

1. **Explicit Base Model Detection:** We first parse the model name for explicit references to base models with size specifications (e.g., Llama-3.1-8B). This is implemented through specialized regular expression patterns tailored to each model family's naming conventions. For instance, Gemma-2-9B is unambiguously matched to the Gemma-2-9B model trained on 8 trillion tokens.

2. **Model Name Pattern Inference:** For models lacking explicit base references, we perform broader pattern matching on model names, scanning for family indicators (e.g., "mistral", "qwen2.5") and version numbers. This layer identifies the model family but may not precisely determine the variant, necessitating parameter-based disambiguation in some cases. For example, detecting "llama-3" in the name identifies the family but requires parameter count verification to distinguish between 8B and 70B variants.

3. **Architecture-based Attribution:** Lastly, we leverage architecture information combined with parameter counts. This approach varies by model family:
   - For Llama models, we employ stringent parameter matching (e.g., 7.8-8.3B for Llama-3-8B) to prevent false positives, as many models adopt the Llama architecture without using Llama weights.
   - For other architectures (e.g., Mistral, Qwen), we implement more generous parameter ranges and higher confidence attribution, as architecture adoption typically indicates weight inheritance.
   - Size-variant mapping is crucial for families like Gemma-2, where pretraining compute differs by size (2B: 2T tokens, 9B: 8T tokens, 27B: 13T tokens).

The algorithm traverses these layers sequentially, defaulting to the highest-confidence identification available. When all layers fail to produce a sufficient confidence match, the algorithm returns null rather than making low-confidence attributions. This ensures precision over recall, maintaining the reliability of identified mappings. Upon successful model identification, we retrieve the corresponding pretraining token count from our comprehensive knowledge base, which consolidates information from research papers, technical reports, and official documentation. This multi-layered approach

## C  DETAILS OF HCA AND ITS THEORETICAL GUARANTEE

### C.1  HCA: HIERARCHICAL COMPONENT ANALYSIS

In Jin & Syrgkanis (2024), the authors introduced the LiNGCReL algorithm identifiability guarantees in Theorem 1 for *exact* SCMs. Here we introduce hierarchical Component Analysis (HCA) that is equivalent to LiNGCReL in the exact setting, but with several modifications to make it fit into our context.

The first step, same as Jin & Syrgkanis (2024), is to apply linear ICA to each individual domain. Recall that ICA's goal is to the independent signals; in our setting, it recovers the *ICA unmixing matrix* $M_k$ that maps observed $x$ to the source variables $\epsilon^{(k)}$ defined in Equation (1). This shall be carefully distincted from $H = (G^\top G)^{-1} G^\top$ which is the unmixing matrix for CRL. When the SCM is exact, we would have $P_k M_k = B_k H$, where $P_k$ is some permutation matrix. The main challenge of CRL is that we only know $M_k = B_k H, k \in [K]$ but each $P_k$ is unknown.

The second and main part of our algorithm is presented in Algorithm 2. The algorithm is motivated by the observation that, since the unmixing matrix $H$ is the same across all domains, the structure of any row spaces of $B_k, k \in [K]$, which are unknown, is captured by the row structures of the known ICA unmixing matrices $B_k H$. Moreover, given an already-recovered subgraph $\mathcal{G}_1$ of $\mathcal{G}$, one can discover some $v \notin \mathcal{G}_1$ such that $\mathrm{pa}_{\mathcal{G}}(v) \subseteq \mathcal{G}_1$, if the corresponding rows in each $B_k$, after projecting onto the row spaces corresponding to the the orthogonal complement of the row space of already-recovered nodes, is rank-1. This is because this rank captures the "remaining degree of freedom" of $v$ conditioned on $\mathcal{G}_1$, which equals one if and only if all its parents are in $\mathcal{G}_1$.

While this idea is close to the original LiNGCReL, some key differences are worth-noticing:

1. Compared with LiNGCReL, HCA only recovers a transitive closure $\bar{\mathcal{G}}$ of the true graph $\mathcal{G}$[3]. It is still possible to infer whether each edge in $\bar{\mathcal{G}}$ indeed exists in $\mathcal{G}$ (see appendix). For simplicity and due to the fundamental inexactness of our model, we do not perform this step here. Equivalently, we are only imposing the constraint that each $B_k$ is upper-triangular, without assuming that any other entries are also zero.

2. The identifiability guarantee of LiNGCReL makes the restrictive assumption that the distribution $\epsilon^{(k)}$ does not depend on $k$. This assumption is indeed unnecessary; the price to pay is a more complicated approach to identify the "correct matching" between the components of $\epsilon^{(k)}$. This step could be computationally expensive, but works well in our context where $d$ is small.

3. We determine the matrices $B_k, k \in [K]$ by explicitly optimizing the distance between the recovered unmixing matrix and the target unmixing matrix. Compared with LiNGCReL that sets the entries of $B_k$'s as the projection coefficients in Algorithm 1, which is theoretically equivalent for exact causal models, this extra step provides additional flexibility that optimizes the fitting quality in the presence of inexactness.

---

**Algorithm 1:** `Ortho-proj`$(S, \{A_k\}_{k=1}^K)$

**Input:** Ordered set $S = \{s_1, s_2, \ldots, s_m\} \subseteq [d]$, index $i \notin S$, $A_k \in \mathbb{R}^{d \times n}$ for $k \in [K]$.
**Output:** Residual matrices $\{R_k\}_{k=1}^K$.
**for** $k \leftarrow 1$ **to** $K$ **do**
$\quad W \leftarrow \mathrm{span}\{(A_k)_s : s \in S\};$ `// (`$A_k$`)`$_s$ `is the s-th row of` $A_k$
$\quad R_k \leftarrow \mathrm{proj}_{W^\perp}(A_k);$`// Row-wise orthogonal projection`
**end**

---

[3]The transitive closure of a directed acyclic graph (DAG) $\mathcal{G}$ is obtained by drawing an edge $i \to j$ for any $i$ and $j$ such that $i$ is an ancestor in $j$, *i.e.*, there is a path $i = i_0 \to i_1 \to \cdots \to i_k = j$ in $\mathcal{G}$.

---

**Algorithm 2:** `Hierarchical component analysis`

**Input:** Matrices $\boldsymbol{M}_k \in \mathbb{R}^{d \times n}$, $k \in [K]$.
**Output:** The optimal unmixing matrix $\hat{\boldsymbol{H}}^*$ and weight matrices $\{\hat{\boldsymbol{B}}_k^*\}_{k=1}^K$.
Let $S_d$ be the set of all permutations of $\{1, 2, \ldots, d\}$. min_mic_score $\leftarrow \infty$ ;
$\hat{\boldsymbol{H}}^* \leftarrow$ null; $\quad \{\hat{\boldsymbol{B}}_k^*\}_{k=1}^K \leftarrow$ null;
**for** *each permutation combination $\pi = (\pi_1, \ldots, \pi_K) \in (S_d)^K$* **do**
    `// 1.  Apply the current permutation to each matrix`
    Let $\boldsymbol{M}_k'$ be the matrix $\boldsymbol{M}_k$ with rows permuted according to $\pi_k$, for $k = 1, \ldots, K$. ;
    `// 2.  Generate candidate` $\hat{\boldsymbol{H}}_\pi$ `based on permuted matrices`
    **for** $j \leftarrow 0$ **to** $d-1$ **do**
        $S_{ortho} \leftarrow \{j+1, j+2, \ldots, d\}$;
        $\{\boldsymbol{R}_k'\}_{k=1}^K \leftarrow$ `Ortho-proj`$(S_{ortho}, \{\boldsymbol{M}_k'\}_{k=1}^K)$;
        `// Extract principal direction from the (j+1)-th rows of`
        `   residuals`
        $\tilde{\boldsymbol{R}} \leftarrow [(\boldsymbol{R}_1')_{j+1}^\top, \ldots, (\boldsymbol{R}_K')_{j+1}^\top]^\top$; `// Stack the (j+1)-th rows`
        $\boldsymbol{h}_{j+1}' \leftarrow \boldsymbol{v}_1(\tilde{\boldsymbol{R}})$; `// Top right singular vector`
    **end**
    $\hat{\boldsymbol{H}}_\pi \leftarrow [\boldsymbol{h}_1', \ldots, \boldsymbol{h}_d']^\top$; `// Construct candidate H. Optionally:`
        `Gram-Schmidt`$(\boldsymbol{h}_1', \ldots, \boldsymbol{h}_d')$
    `// 3.  Compute optimal upper-triangular` $\hat{\boldsymbol{B}}_{k,\pi}$
    Let $\mathcal{T}(d)$ be the set of $d \times d$ upper-triangular matrices. ;
    **for** $k \leftarrow 1$ **to** $K$ **do**
        $\hat{\boldsymbol{B}}_{k,\pi} \leftarrow \arg\min_{\boldsymbol{B} \in \mathcal{T}(d)} \|\boldsymbol{M}_k' - \boldsymbol{B}\hat{\boldsymbol{H}}_\pi\|_F^2$; `// Best upper-triangular`
        `  estimate`
    **end**
    `// 4.  Compute the MIC score for this permutation using`
    `   Proposition 1`
    current_mic_score $\leftarrow$ ComputeMIC$(\{\boldsymbol{M}_k'\}_{k=1}^K, \{\hat{\boldsymbol{B}}_{k,\pi}\}_{k=1}^K, \hat{\boldsymbol{H}}_\pi)$ ;
    `// 5.  Update if this is the best score found so far`
    **if** *current_mic_score < min_mic_score* **then**
        min_mic_score $\leftarrow$ current_mic_score ;
        $\hat{\boldsymbol{H}}^*, \{\hat{\boldsymbol{B}}_k^*\}_{k=1}^K \leftarrow \hat{\boldsymbol{H}}_\pi, \{\hat{\boldsymbol{B}}_{k,\pi}\}_{k=1}^K$ ;
    **end**
**end**
**return** $\hat{\boldsymbol{H}}^*, \{\hat{\boldsymbol{B}}_k^*\}_{k=1}^K$

---

## C.2 IDENTIFIABILITY GUARANTEE FOR HCA

In this subsection, we provide our main identifiability result for HCA in the special case when the graph $\mathcal{G}$ is known to be a complate DAG with $i \to j$ for all $i < j$. Equivalently, this means that each $\boldsymbol{A}_k$ is lower-triangular.

**Assumption 1.** *(Node-level non-degeneracy, adapted from (Jin & Syrgkanis, 2024, Assumption 5))* *We assume that the matrices $\{\boldsymbol{B}_k\}_{k=1}^K$ are node-level non-degenerate, i.e., for all node $i \in [d]$, we have* $\dim \text{span} \langle (\boldsymbol{B}_k)_i : k \in [K] \rangle = |\text{pa}_{\mathcal{G}}(i)| + 1$, *where $(\boldsymbol{B}_k)_i$ is the $i$-th row of $\boldsymbol{B}_k$.*

As shown in Jin & Syrgkanis (2024), this assumption holds as long as the $K$ weight vectors at node $i$ across $K$ domains do not lie in a low-dimensional vector space, which generally holds. To ensure identifiability, we also require that the components of noise variables are non-Gaussian and have different distributions.

**Assumption 2.** *For all $k \in [K]$, each component of $\boldsymbol{\epsilon}^{(k)}$ follows a different distribution, and all of them are non-Gaussian.*

**Remark 2.** *With a more involved procedure, Jin & Syrgkanis (2024) showed that one can identify $z_i, i \in [d]$ up to a "surrounding node ambiguity" in the case when $\mathcal{G}$ is unknown. Specifically, this means that the $\mathcal{G}$ can be fully recovered and the identified factor $z_i'$ is some linear combination of $z_j$'s with $j \in \text{sur}_{\mathcal{G}}(i) := \{i\} \cup \{i' \in \text{pa}_{\mathcal{G}}(i) : \text{ch}_{\mathcal{G}}(i) \subseteq \text{ch}_{\mathcal{G}}(i')\}$. Moreover, this ambiguity is intrinsic in this setting.*

Our main result is stated below:

**Theorem 1.** *Suppose that $K \geq d$, then if the ICA step is exact, one can recover the mixing matrix $\boldsymbol{G}$ up to a left multiplication of lower-triangular matrix. Equivalently, it recovers latent factors $z'_1, \cdots, z'_d$ where $z'_i$ is a linear mixture of the true latent factors $z_j, j < i$.*

The remaining part of this subsection is devoted to proving Theorem 1.

By our assumption of the causal model, we know that in the $k$-th domain, the observations and the noise variables are related via $\boldsymbol{\epsilon}^{(k)} = \boldsymbol{B}_k \boldsymbol{H} \boldsymbol{x}$. Since we assume that the ICA is exact, the uniqueness of ICA in the non-Gaussian setting (Eriksson & Koivunen, 2004) implies that the umixing matrix that leads to independent source variables must be unique up to row permutations. In other words, there exists some permutation matrix $\boldsymbol{P}_k^*$, such that $\boldsymbol{P}_k^* \boldsymbol{M}_k = \boldsymbol{B}_k \boldsymbol{H}, \forall k \in [K]$. Without loss of generality, we also assume that $\boldsymbol{H}$ is orthonormal, since otherwise one can always consider a QR factorization $\boldsymbol{H} = \boldsymbol{U} \tilde{\boldsymbol{H}}$ where $\boldsymbol{U}$ is lower-triangular and $\tilde{\boldsymbol{H}}$ is orthonormal, and one can treat $\boldsymbol{B}_k \boldsymbol{U}$ as the new $\boldsymbol{B}_k$.

Recall that our algorithm goes through all possibilities of permutations $\boldsymbol{P}_k, k \in [K]$ and pick one with the smallest MIC. To begin with, it is not hard to see the following fact:

**Proposition 2.** *Suppose that $\boldsymbol{M}'_k = \boldsymbol{P}_k^* \boldsymbol{M}_k$, then running the subroutine in Algorithm 2 on $\boldsymbol{M}'_k, k \in [K]$ would give a zero MIC.*

*Proof.* Recall that $\boldsymbol{M}'_k = \boldsymbol{B}_k \boldsymbol{H}$. We will prove by induction that each row $\boldsymbol{h}'_i$ of the recovered matrix $\hat{\boldsymbol{H}}_\pi$ in Algorithm 2 is parallel to $\boldsymbol{h}_i$ (*).

For $i = 1$, since $\boldsymbol{B}_k$ is lower-triangular, and its diagonal entries $\boldsymbol{\Omega}_k^{-1/2}$ are nonzero, so the last rows of $\boldsymbol{B}_k \boldsymbol{H}, k \in [K]$ is a nonzero multiple of $\boldsymbol{h}_1$. By definition, $\boldsymbol{h}'_1$ is the principal component of these rows, which is obviously parallel to $\boldsymbol{h}_1$.

Suppose the conclusion holds for all $j < j_0$, we now prove it for $j = j_0$. Since $\boldsymbol{B}_k$ is lower-triangular, the induction hypothesis implies that for each $k \in [K]$, $\text{span}\langle (\boldsymbol{M}'_k)_i : i < j \rangle = \text{span}\langle \boldsymbol{h}_i : i < j \rangle$. By definition, $(\boldsymbol{R}_k)_j$ is the orthogonal projection of $(\boldsymbol{M}_k)_j$ onto this subspace. Notice that $(\boldsymbol{M}_k)_j \in \text{span}\langle \boldsymbol{h}_i : i \leq j \rangle$, is then easy to see that this projection is nothing but a constant multiple of $\boldsymbol{h}_j$, since this is the unique direction in $\text{span}\langle \boldsymbol{h}_i : i \leq j \rangle$ that is orthogonal to $\text{span}\langle \boldsymbol{h}_i : i \leq j \rangle$. Hence by definition, we have $\boldsymbol{h}'_j = \alpha_j \boldsymbol{h}_j$ for some scalar $\alpha_j$. This concludes the proof of (*).

From (*), it is easy to see that the best lower-triangular estimate $\hat{\boldsymbol{B}}_{k,\pi}$ is equal to $\boldsymbol{B}$ up to some row-wise scaling, and that $\hat{\boldsymbol{B}}_{k,\pi} \hat{\boldsymbol{H}}_\pi = \boldsymbol{M}'_k$. Hence the MIC is zero by definition. $\square$

To complete the proof of Theorem 1, we need to show that any permutation that achieves a zero MIC successfully recovers the causal graph up to transitive closure. Specifically, suppose that some permutation matrices $\boldsymbol{P}_k, k \in [K]$ leads to a zero MIC, let $\boldsymbol{Q}_k = \boldsymbol{P}_k \boldsymbol{P}_k^*$, then $\boldsymbol{M}_k = \boldsymbol{Q}_k \boldsymbol{B}_k \boldsymbol{H}$. We show that

1. $\boldsymbol{Q}_1 = \boldsymbol{Q}_2 = \cdots = \boldsymbol{Q}_d$, and

2. Suppose that the $j$-th row of $\boldsymbol{Q}_1$ is $\boldsymbol{e}_{i_j}$, then $i_1, i_2, \cdots, i_d$ is a topological ordering of the graph $\mathcal{G}$, meaning that $\text{pa}_\mathcal{G}(i_j) \subseteq \{i_1, \cdots, i_{j-1}\}$.

We say that a row index $j$ is "good" if the $j$-th row of $\boldsymbol{Q}_k, k \in [K]$ are equal and the second condition above is satisfied up to $j$ (*i.e.* $i_1, \cdots, i_j$ is an ancestral set of $\mathcal{G}$), and is "bad" otherwise. Then it suffices to show that all $j \in [d]$ are good.

Suppose the contrary holds, let $j = j_0$ be the smallest bad index. A zero MIC implies that $k \in [K]$,

$$\left[ \sum_{i=1}^j (\hat{\boldsymbol{B}}_k)_{ji} \hat{\boldsymbol{h}}_i \right] \boldsymbol{M}_k^\top (\boldsymbol{M}_k \boldsymbol{M}_k^\top)^{-1} = \lambda_{kj} \boldsymbol{e}_j.$$

Hence,

$$\left[ \sum_{i=1}^j (\hat{\boldsymbol{B}}_k)_{ji} \hat{\boldsymbol{h}}_i - \lambda_{kj} (\boldsymbol{M}_k)_j \right] \boldsymbol{M}_k^\top (\boldsymbol{M}_k \boldsymbol{M}_k^\top)^{-1} = 0 \quad \Rightarrow \quad \sum_{i=1}^j (\hat{\boldsymbol{B}}_k)_{ji} \hat{\boldsymbol{h}}_i - \lambda_{kj} (\boldsymbol{M}_k)_j \in V^\perp,$$

$$(2)$$

where $V$ is the row space of $\boldsymbol{H}$. The last step holds since the row space of $\boldsymbol{M}_k$ is also $V$. However, by induction hypothesis, the first $(j-1)$ rows of $\boldsymbol{M}_k$ are exactly the $i_1, i_2, \cdots, i_{j-1}$-th rows of $\boldsymbol{B}_k \boldsymbol{H}$. The construction of $\hat{\boldsymbol{H}}$ and $\hat{\boldsymbol{B}}_k$ imply that the $s$-th $(s < j)$ row of $\hat{\boldsymbol{H}}$ is equal to the $i_s$-th row of $\boldsymbol{H}$, and the first $(j-1)$ rows of $\hat{\boldsymbol{B}}_k$ are equal to the $i_1, i_2, \cdots, i_{j-1}$-th rows of $\boldsymbol{B}_k$. Let $V_i = \operatorname{span}\langle \hat{\boldsymbol{h}}_1, \cdots, \hat{\boldsymbol{h}}_i \rangle$, then we have that

$$\operatorname{proj}_{V_{j-1}^\perp}\left(\sum_{i=1}^{j}(\hat{\boldsymbol{B}}_k)_{ji}\hat{\boldsymbol{h}}_i - \lambda_{kj}(\boldsymbol{M}_k)_j\right) = \operatorname{proj}_{V_{j-1}^\perp}\left((\hat{\boldsymbol{B}}_k)_{jj}\hat{\boldsymbol{h}}_j - \lambda_{kj}(\boldsymbol{M}_k)_j\right)$$
$$= (\hat{\boldsymbol{B}}_k)_{jj}\operatorname{proj}_{V_{j-1}^\perp}(\hat{\boldsymbol{h}}_j) - \lambda_{kj}\operatorname{proj}_{V_{j-1}^\perp}(\boldsymbol{M}_k)_j. \tag{3}$$

However, equation 2 implies that this quantity is equal to zero. As a result, we have

$$\operatorname{rank}\left\langle \operatorname{proj}_{V_{j-1}^\perp}(\boldsymbol{M}_k)_j : k \in [K] \right\rangle = 1.$$

Let the $j$-th row of $\boldsymbol{Q}_k$ be $\boldsymbol{e}_{j_k}, k \in [K]$. Then the above equation becomes

$$\operatorname{rank}\left\langle \operatorname{proj}_{V_{j-1}^\perp}(\boldsymbol{B}_k\boldsymbol{H})_{j_k} : k \in [K] \right\rangle = 1. \tag{4}$$

In the following, we show that this property can only hold when $j$ is good. Note that

$$\operatorname{proj}_{V_{j-1}^\perp}(\boldsymbol{B}_k\boldsymbol{H})_{j_k} = \sum_{i \in \bar{\operatorname{pa}}_\mathcal{G}(j_k) \setminus \{i_1, \cdots, i_{j-1}\}} (\boldsymbol{B}_k)_{j_k,i}\operatorname{proj}_{V_{j-1}^\perp}(\boldsymbol{h}_i), \tag{5}$$

where $\bar{\operatorname{pa}}_\mathcal{G}(i) = \operatorname{pa}_\mathcal{G}(i) \cup \{i\}$. For $k \neq l$, equation 4 implies that $\operatorname{proj}_{V_{j-1}^\perp}(\boldsymbol{B}_k\boldsymbol{H})_{j_l}$ and $\operatorname{proj}_{V_{j-1}^\perp}(\boldsymbol{B}_k\boldsymbol{H})_{j_l}$ are colinear, but since $\boldsymbol{H}$ has full row rank, $\operatorname{proj}_{V_{j-1}^\perp}(\boldsymbol{h}_i), i \in [d] \setminus \{i_1, \cdots, i_{j-1}\}$ are independent, so we must have $\bar{\operatorname{pa}}_\mathcal{G}(j_k) \setminus \{i_1, \cdots, i_{j-1}\} = \bar{\operatorname{pa}}_\mathcal{G}(j_l) \setminus \{i_1, \cdots, i_{j-1}\}$. In particular, $j_k \in \operatorname{pa}_\mathcal{G}(j_l)$ and $j_l \in \operatorname{pa}_\mathcal{G}(j_k)$, so we must have $j_k = j_l$. Thus $j_1 = j_2 = \cdots = j_K$.

By Assumption 2,
$$\operatorname{rank}\langle(\boldsymbol{B}_k\boldsymbol{H})_{j_1} : k \in [K]\rangle = \left|\operatorname{pa}_\mathcal{G}(j_1)\right| + 1,$$

so that

$$\operatorname{rank}\left\langle \operatorname{proj}_{V_{j-1}^\perp}(\boldsymbol{B}_k\boldsymbol{H})_{j_k} : k \in [K] \right\rangle \geq \left|\operatorname{pa}_\mathcal{G}(j_1)\right| + 1 - \left|\operatorname{pa}_\mathcal{G}(j_1) \cap \{i_1, \cdots, i_{j-1}\}\right|.$$

Hence it mus be the case that $\operatorname{pa}_\mathcal{G}(j_1) \subseteq \cap\{i_1, \cdots, i_{j-1}\}$, concluding the proof.

# D ADDITIONAL PCA ANALYSES

In this section, we examine different ways to choose the domains from the open LM leaderboard and discuss our findings.

**Domain-specific PCA.** While Figure 2a indicates that the complete leaderboard dataset is approximately low-rank, this global characteristic does not inherently imply a similar low-rank structure for benchmark performance data within individual domains. It is plausible that some domains possess full-rank data, but these higher-rank properties are obscured or averaged out when the entire leaderboard is considered. To investigate this, we performed PCA on the eight domains containing the largest number of model entries. As illustrated by the analysis of their leading principal components in Figure 9, all examined domains are effectively rank-3, with the exception of Gemma-2-9B, which exhibits an approximate rank of 2.

**PCA for more base models.** We first provide an extended version of Figure 2c for 20 most frequently used base models of the open LM leaderboard.

**Mixture of Experts (MoE).** We investigate the MoE architecture, which is used by Mixtral, and more recently, by Deepseek. The information of whether a model uses the MoE architecture is directly available from our leaderboard. In Figure 11a, we plot the principal component subspace distances between MoE models and non-MoE models. We also include two architectures upon which a vast

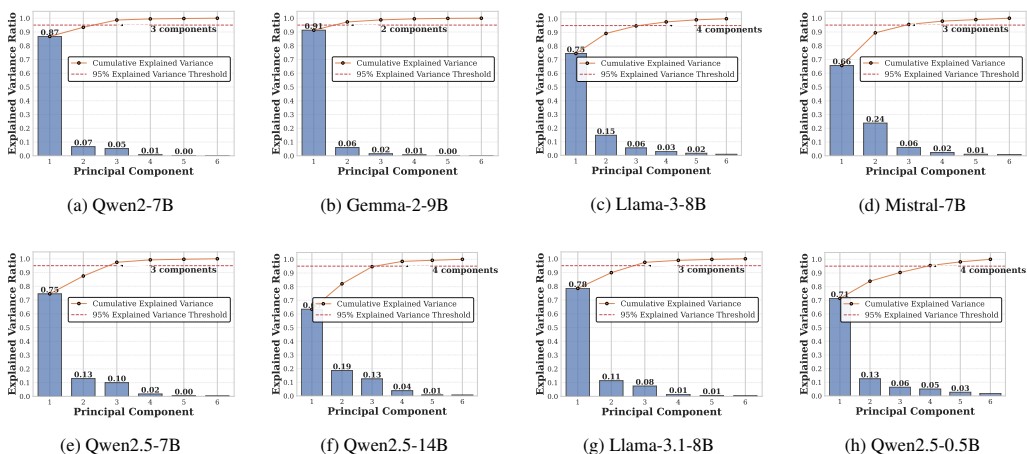

Figure 9: Results for running PCA on individual domains.

majority of MoE models are built. We can see that there is little difference in the principal component subspaces.

**Different relative sizes of $N$ and $D$.** While the pretraining compute $C \approx 6ND$ is well-known to directly affect the model performance, the precise roles of $N$ and $D$ remain unclear. Our PCA results in Figure 11b considers four domains of data that contain models with small $N$ and large $D$, large $N$ and small $D$, small $N$ and small $D$, large $N$ and large $D$ respectively. The finding is intriguing – it shows that the small $N$, large $D$ domain has a principal component subspace that is quite different from the other domains. Further investigation by controlling for base models show that this is just a coincidence, as shown in Figure 11c. In this figure, we consider domains corresponding to the two most frequent base models for each domain used in Figure 11b. We find that while three principal component subspaces in Figure 11b look similar, they are actually the mixture of domains with very different principal component subspaces. This further hightlights the importance of controlling for the base model in causal analysis, as in the approach of our main work.

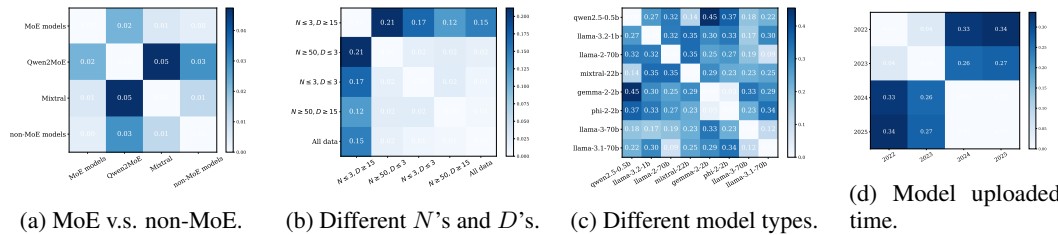

Figure 11: PCA Results comparing principal component subspaces for different criteria.

**Different uploaded time.** Lastly, we define domains according to which year the model is uploaded. In Figure 11d, we find that there is a clear watershed between 2023 and 2024. Similar findings are also made in Dominguez-Olmedo et al. (2025), where the authors argue that after November 2023, "training on test task" becomes more prevalent.

It should be noticed that similarity of PC subspaces is a necessary but not sufficient conditions for our causal analysis. Domains with similar PC subspaces may not be explained by a linear causal model. Moreover, we apply causal analysis to domains defined by base models primarily because this helps us remove all confounders related to the pretraining stage. On this other hand, difference in PC subspaces likely indicate some heterogeneous causal patterns. We leave the analyses of these patterns to future work.

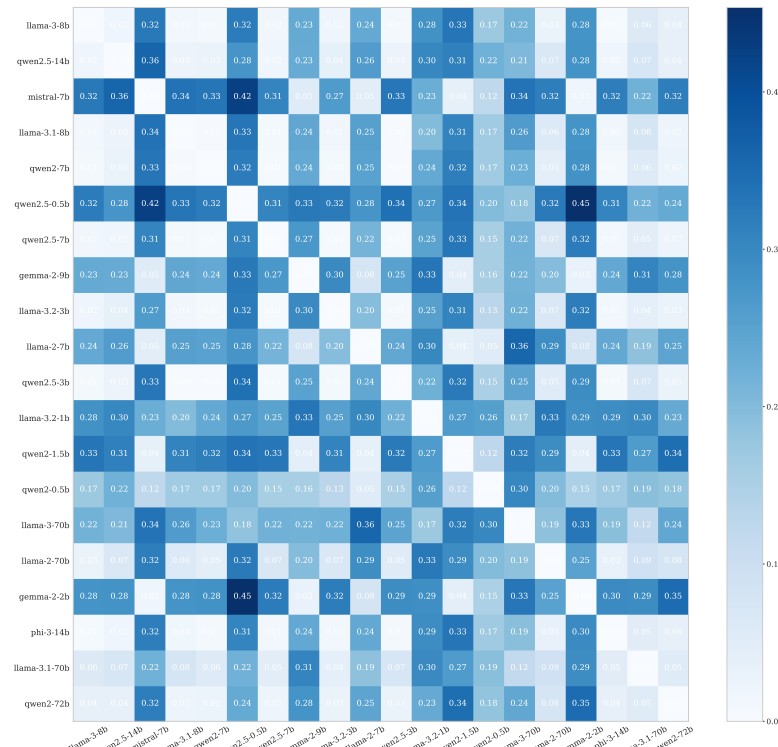

Figure 10: Pairwise cosine distance matrix for 20 base models.

# E  SCALING LAWS AND THE EFFECT OF FINE-TUNING

Existing predictive models for language model performances are typically restricted to pretrained models. This is not unexpected, since it is hard to characterize the performance gains in post training in terms of the relevant factors. In this section, we point out some of the key challenges in understanding the effect of fine-tuning.

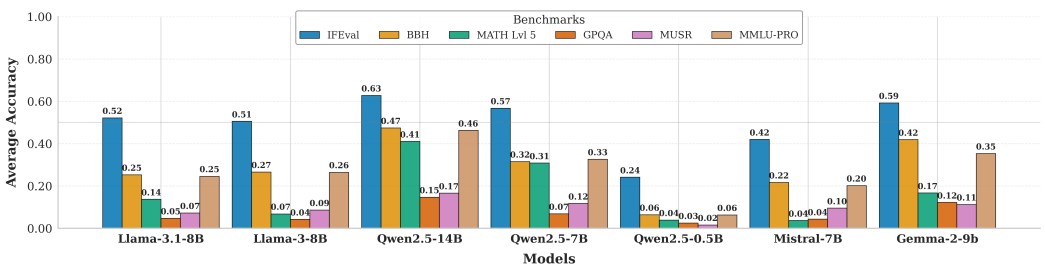

Figure 12: The average benchmark performance of fine-tuned models on the open LM leaderboard with three base models in different sizes.

As illustrated in Figure 12, models fine-tuned on more powerful base models tend to exhibit uniformly better performance across all benchmarks. In other words, base model is a common confounder of all benchmark performances. We observe that base model also confounds the amount of improvment one can achieve on all benchmarks. To illustrate this point, we estimate the average treatment effect (ATE) of $T$ on all six benchmarks of the open LM leaderboard using the backdoor adjustment formula $\mathbb{E}[Y \mid do(T)] = \int \mathbb{E}[Y \mid T, X = x] p_X(x) \mathrm{d}x$, where $X = \log(C)$ is the log pretraining compute and $p_X(\cdot)$ its density. As illustrated in Figure 13, fine-tuning yields substantial gains on math reasoning and instruction-following benchmarks, while producing little to negative change on general reasoning and QA-based tasks. Examining Llama- versus Qwen-based variants separately, we observe that Qwen models gain more from fine-tuning on math reasoning and instruction-following, yet incur larger drops on general reasoning.

**Remark 3.** *Caution is warranted when interpreting the causal implications of the estimates presented in Figure 13. These values represent true Average Treatment Effects (ATEs) only when the conditional ignorability assumption—fundamental to causal inference—is satisfied. In our context, this assumption requires that different base models experience equivalent "distributions of interventions" across tasks. For example, if Qwen demonstrates superior performance gains compared to Llama on mathematics-related tasks, conditional ignorability would be violated if researchers strategically selected Qwen models more frequently for mathematical applications to maximize performance outcomes. We contend that this assumption is difficult to substantiate in practice. An important open research question remains: how might we circumvent this methodological challenge when limited to observational performance data? Developing robust approaches that account for such selection biases represents a significant opportunity for future work in this domain.*

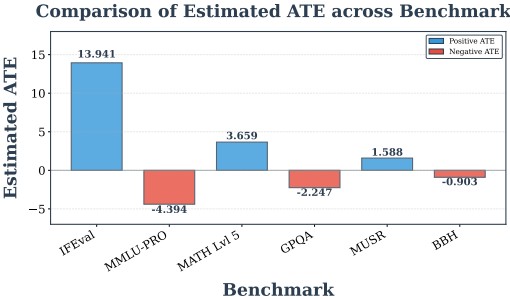

Figure 13: Estimates of the average effect of fine-tuning.

### E.1 HETEROGENEITY OF FINE-TUNING EFFECTS

Scaling law (Kaplan et al., 2020) has been widely adopted to predict the benchmark performance from pretraining compute. Later, (Ruan et al., 2024; Ren et al., 2025) used sigmoid scaling laws to fit the principal components of performance data from multiple benchmarks. Could scaling law alone explain the leaderboard data? To investigate this question, we let $C \approx 6 \cdot N \cdot D$ be the pretraining compute, and fit a sigmoid regression equation $Y \approx \frac{L}{1+\exp(-k(\log C - \log C_0))} + \tau T + b$, where $L, k, C_0, b, \tau$ are unknown parameters, and $T$ is a binary treatment variable indicating whether a model is fine-tuned or pretrained.

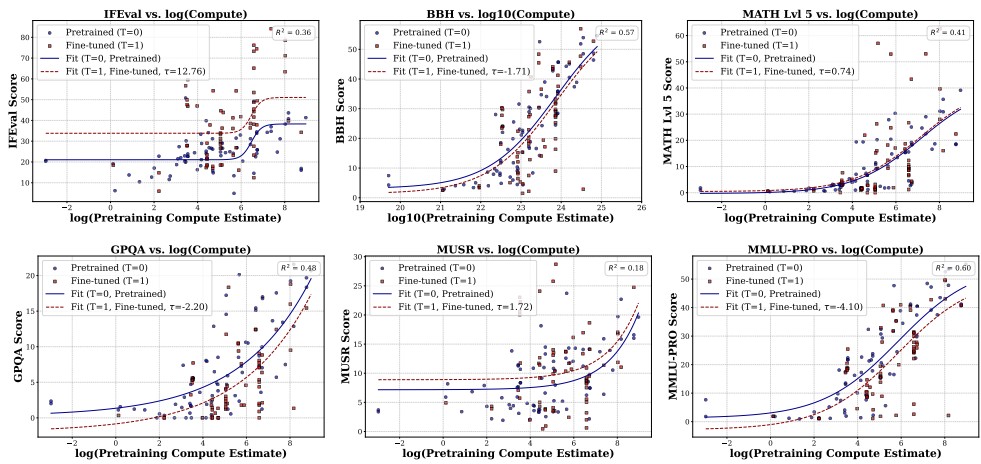

Figure 14: Sigmoid scaling laws of benchmark accuracies for pretrained and fine-tuned models. Top row: all pretrained and fine-tuned models. Middle row: Llama-based models only. Bottom row: Qwen-based models only.

We fitted a sigmoid curve to the benchmark results of all officially released models on the leaderboard (see Figure 14). Our findings indicate that scaling laws more faithfully describe trends on BBH, MMLU-Pro and GPQA—than the remaining benchmarks. Our conjecture is that the former three benchmarks are more "knowledge-driven", in the sense that many questions in these benchmarks merely test whether the model possesses cetain knowledge. As a result, fine-tuning, mainly focusing on reasoning and alignment, can being negligible effect. By contrast, performances on tasks requiring other proficiencies (e.g. instruction following in IFEval, mathematical reasoning in MATH or multi-step soft reasoning in MUSR) are much easier to improve by fine-tuning.

## F    DETAILS FOR MATRIX COMPLETION

In this subsection, we provide a detailed description of the experimental setup in Remark 1. Specifically, our goal is to show how to accurately impute missing leaderboard data when the benchmark performances of LMs are only partially observed. Indeed, this task can be naturally viewed as an instance of matrix completion, where $\boldsymbol{X} \in \mathbb{R}^{N \times d}$ is the performance matrix for $N$ models and $d = 6$ benchmarks, with missing entries.

Restricting ourselves to the missing entries in one particular domain – the group of models fine-tuned on Qwen2.5-14B – we consider a "global" and a "local" approach to perform matrix completion. In the global approach, we apply nuclear norm regularization (NNR, Recht (2011)) to the whole leaderboard data $\mathcal{D} \in \mathbb{R}^{N \times d}$, while the local approach only runs NNR on the submatrix $\mathcal{D}_2$ that only contains rows in $\mathcal{I}_2$, following the notation in Section 1.1.

We conduct synthetic experiments to simulate two different scenarios. First, for the case when the benchmark accuracies are missing at random, we remove each entry of $\mathcal{X}$ independently with probability $p = 0.8$, as visualized in Figure 3 (a). Second, we consider a "block" missing pattern as visualized in Figure 3 (b), where performance on two benchmarks are fully observed, while for the ramaining four benchmarks, the performance data for a $p = 0.1, 0.2, \cdots, 0.9$ fraction of models is missing. We repeat the experiment 1000 times for the first case, and for all $\binom{6}{3} = 20$ possible sets of fully observed benchmarks of size 3 for the second case. Since standard NNR does not perform well on block missing entries, we use structured matrix completion that is designed specifically for handling this case (Cai et al., 2016). The RMSEs of the global and local approaches for these two cases are plotted in Figure 3. We can see that for the first case, the local approach is significantly more accurate than the global approach, despite the fact that it relies on fewer rows. For the second case, the local approach also performs better on average.

In Figure 15 we further plot the RMSEs for all 20 possible choices of fully observed columns. The remaining $6 - 3 = 3$ columns have rows that are missing with $p = 0.5$ probability. We observe that the local approach is always no worse than the global one, and in most cases, the local aproach leads to significant improvements.

## G    ADDITIONAL EXPERIMENT RESULTS

### G.1    DETAILS FOR THE HCA RECOVERY IN SECTION 4

In this subsection, we provide more details and results for the recovery of the causal model in Section 4. First, we provide the visualization of the full DGP recovered by HCA *before* the OLS adjustment in Figure 16.

The OLS adjustment essentially operates on the columns of the mixing matrix in Figure 16 by subtracting from the $i$-th column some linear combination of the $j, j < i$ columns. We report the $R^2$ for aligning all six benchmarks with the three capability factors in Table 3a. The findings are particularly interesting if we consider what each benchmark is supposed to measure. Specifically, BBH and MMLU-PRO both contain tasks across different domains and are both related to language understanding and general reasoning, which, intuitively, are more fundamental capabilities. IFEval tests a model's ability of answering questions in correct formats, which is built on top of the language understanding ability. Finally, the MATH Lvl 5 benchmark requires models to answer math questions correctly *and* in the correct format, which is the most ad-hoc capability built on all the previous ones. These intuitions precisely align with the hierarchical structure of capability factors that we recover.

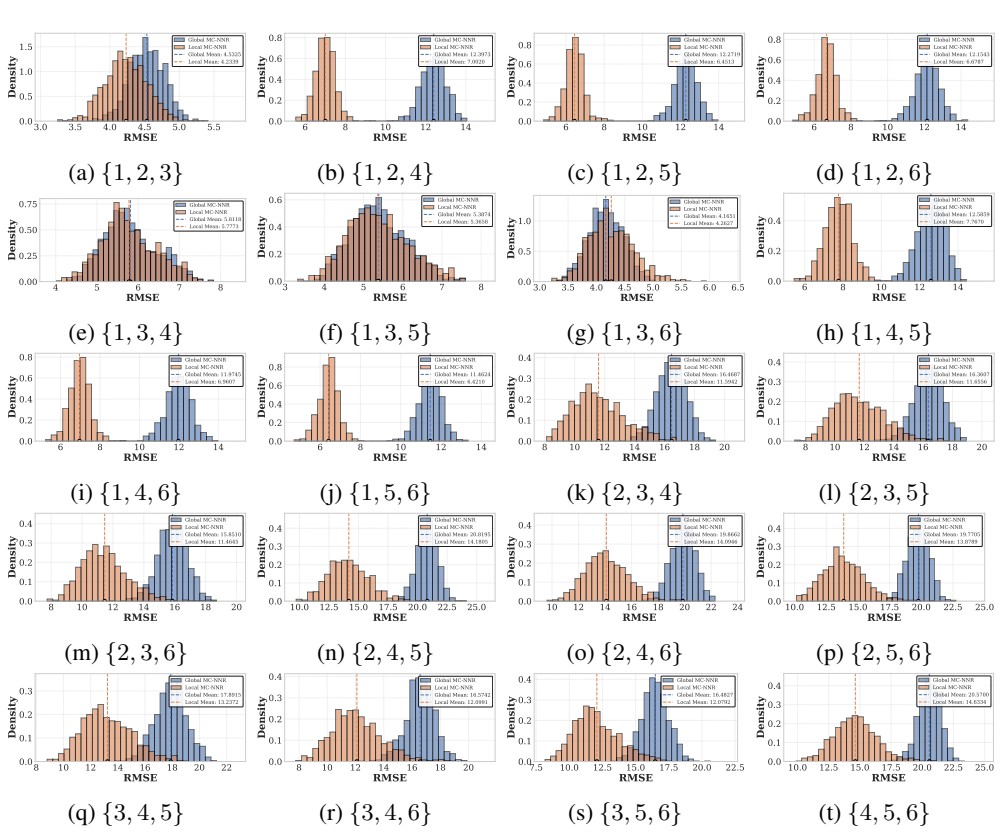

Figure 15: RMSEs of global v.s. local matrix completion for each possible set of fully observed columns. The caption below each figure indicates the columns that are fully observed. Each number representing a benchmark on the leaderboard, with $1, 2, \cdots, 6$ standing for IFEval, BBH, MATH Lvl 5, GPQA, MUSR and MMLU-Pro respectively.

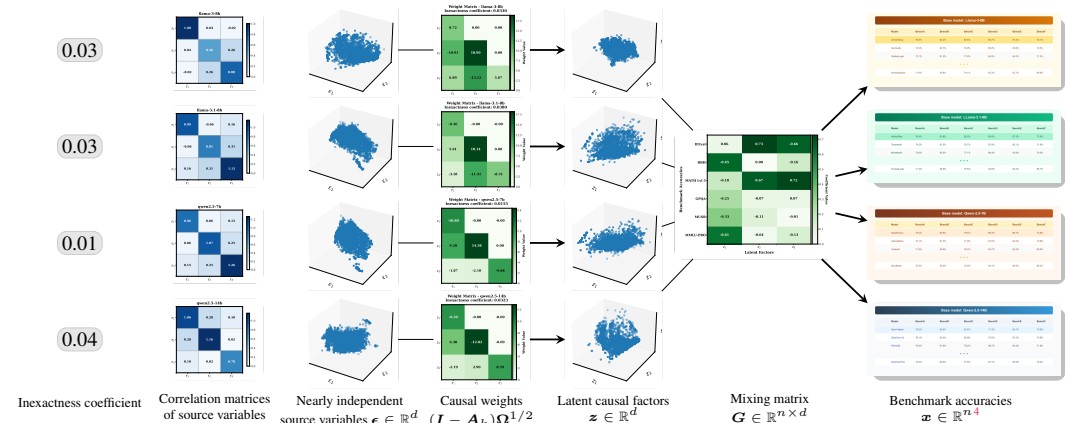

Figure 16: HCA's recovery of the DGP, including the linear SCM (second column) and mixing matrix (fourth column) on four domains (base models): Llama-3-8B, Llama-3.1-8B, Qwen2.5-7B and Qwen2.5-14B. Here, we have $n = 6$ benchmarks, $d = 3$ latent factors and $K = 6$ domains.

More discussions can be found in Section 4.2. A caveat is that this causal structure is only guaranteed to hold for the four base models we consider. As shown in Table 3b, the fitted OLS model can have poor performance on other base models.

|  | IFEval | BBH | MATH | GPQA | MUSR | MMLU-PRO |
|---|---|---|---|---|---|---|
| $z_1$ | 0.36 | **0.96** | 0.56 | 0.73 | 0.57 | **0.96** |
| $z_2$ | **0.92** | 0.53 | 0.66 | 0.57 | 0.58 | 0.54 |
| $z_3$ | **1.00** | 0.43 | **1.00** | 0.18 | 0.14 | 0.16 |

(a) The $R^2$ of running OLS on $z_i$ using $z_j, j > i$ and the benchmark performance as controls.

|  | In Sample | Gemma-2-9B | Mistral-7B | Qwen2.5-0.5B | Qwen2.5-3B | Llama-2-7B | Llama-2-13B | Llama-3.2-1B |
|---|---|---|---|---|---|---|---|---|
| $z_1$ | 0.96 | 0.76 | 0.71 | −0.2 | 0.89 | 0.97 | 0.66 | −0.01 |
| $z_2$ | 0.92 | 0.94 | 0.74 | −1.18 | 0.73 | 0.98 | 0.45 | 0.9 |
| $z_3$ | 1 | 1 | 1 | 0.99 | 1 | 1 | 1 | 1 |

(b) The $R^2$ of the fitted OLS on out-of-sample performance data with different base models.

Table 3: The precise alignment of underlying factors with established benchmarks, coupled with their ability to extend effectively across diverse model domains.

### G.2 COMPLEMENTARY RESULTS FOR SECTION 4

**MIC for all other domain subsets.** In Figure 17, we report the corresponding MIC for all possible choices of domains in $S_{\text{inv}}$. We observe that the four subsets with smallest MIC are achieved by excluding Qwen2-7B.

**Additional metrics for the recovery results.** Note that one potential limitation of MIC is that it is insensitive to the orthogonal complement component of each row in $\hat{B} - \hat{H}$ relative to $M_k$. Therefore, we present two additional metrics indicating how well our causal model fits the observed data, as shown in Table 4. We introduce these metrics since they directly measure how close $\hat{B}_k\hat{H}$ is to the true ICA mixing matrix $M_k$.

**Low-rank approximation error of our causal model.** Recall that we hypothesize that the data is generated from a linear causal model with 3 nodes. This necessarily requires that the performance data across all 6 benchmarks is a matrix with rank at most 3. While we have seen in Figure 2a that this is approximately the case, here we revisit this assumption and see investigate the error induced by this assumption.

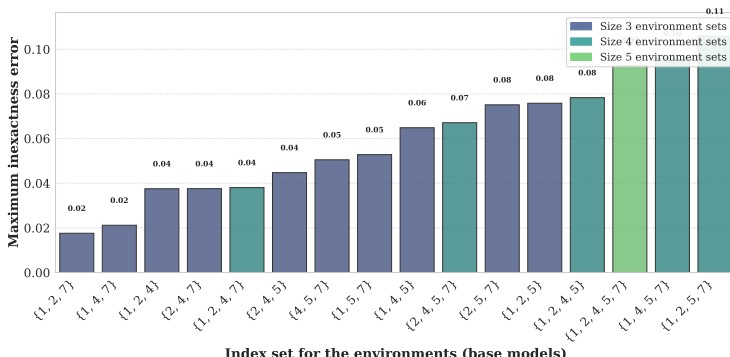

Figure 17: Overview of the MIC obtained by difference choices of domain indices. Here, as we indicated in Section 1.1, indices $1, 2, 4, 5, 7$ correspond to base models Llama-3-8B, Qwen2.5-14B, Llama-3.1-8B, Qwen2-7B and Qwen2.5-7B respectively.

| Node | $z_1$ | $z_2$ | $z_3$ |
|---|---|---|---|
| Rank-1 error | 0.05 | 0.13 | 0.02 |

(a) The amount of variation in $\tilde{R}$ defined in Algorithm 2 uncaptured by a rank-1 matrix in each iteration.

| Domain | Llama-3-8B | Llama-3.1-8B | Qwen2.5-7B | Qwen2.5-14B |
|---|---|---|---|---|
| Unmixing error | 0.17 | 0.20 | 0.16 | 0.23 |

(b) The relative recovery error of the unmixing matrix of ICA for each domain, calculated from $\|M_k - B_k H\|_F / \|M_k\|_F$, where $M_k$ is the ICA unmixing matrix of the $k$-th domain, $B_k$ is the inverse of the recovered weight matrix, and $H$ is the unmixing matrix of CRL.

Table 4: Additional metrics on how good our causal model explains the observed data.

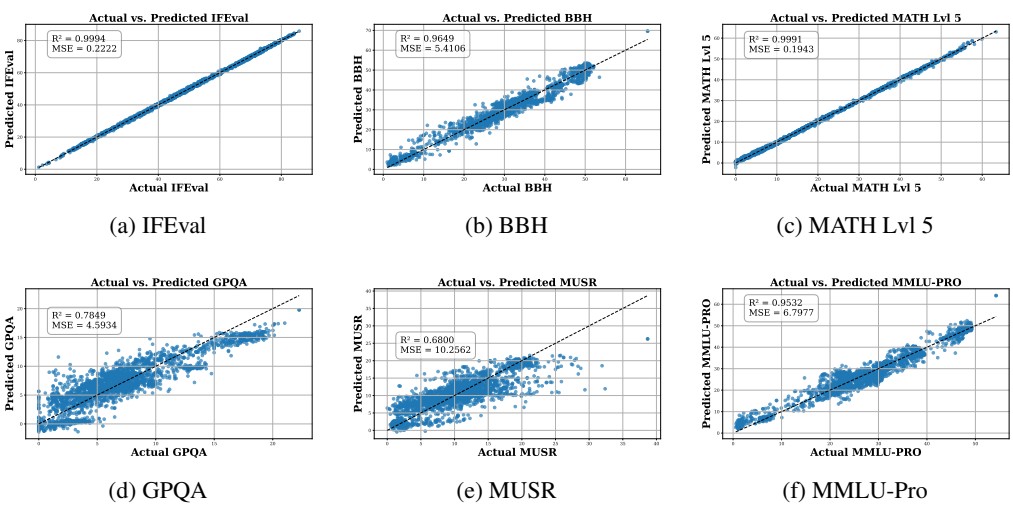

(a) IFEval      (b) BBH      (c) MATH Lvl 5

(d) GPQA      (e) MUSR      (f) MMLU-Pro

Figure 18: Approximation error of the low-rank latent factor space for the observed benchmark performances.

In Figure 18, we plot the approximation errors of the subspace spanned by the values of three latent factors $z_i$, $i = 1, 2, 3$ learned via our algorithm. One can see that the low rank subspace approximates 4 out of 6 benchmarks nearly perfectly. The relatively poor fitting for the remaining two, namely GPQA and MUSR, is partially due to the fact that the model's accuracies on them are systematically lower than the remaining ones. As a result, they would be ignored to some extent when picking the principle components. In terms of the MSE, the error of fitting GPQA is comparable to the remaining four, while that of MUSR is significantly higher.

This highlights a limitation in our current methodology: although we introduce the notion of inexact causal graph for more flexibility, the assumption that each two latent factors have a causal relationship is still restrictive. For instance, it is possible that $z_1, z_2$ are correlated but there exists no causal relationship between them, and both of them causally affect $z_3$. It will be an interesting future direction to investigate how to identify the latent factors in these cases.

### G.3 FILTERING OUT "BADLY" FINE-TUNED MODELS

We notice that some models on the leaderboard are badly fine-tuned, so that their benhmark performances are even worse than the pretrained model. In this subsection, we provide results of our causal analysis with these bad models removed. Removing the bad models allow us to characterize the hierarchical relationship between capabilties that is restricted to "good" fine-tuning strategies. The recovered DGP is shown in Figure 19. After adjusting for the ambiguity as we did in Section 4, we obtain the causal graphs shown in Figure 20. Finally, in Figure 21, we plot the unmixing matrix after adjustment and the relationship between each latent factor and the most indicative benchmark.

The overall pattern that our algorithm discovers is the same as the unfiltered approach. However, we notice that in the filtered case, the MIC is much larger, indicating that the causal model is less well-fitted. This is likely due to the fact that after filtering, the variance of performances on BBH and MMLU-Pro becomes significantly smaller, so that the weights of GPQA and MUSR in $z_1$ are larger compared with the unfiltered case (see Figure 7a). These two benchmarks are relatively not well-explained by our linear causal model, as we discussed in Appendix G.2.

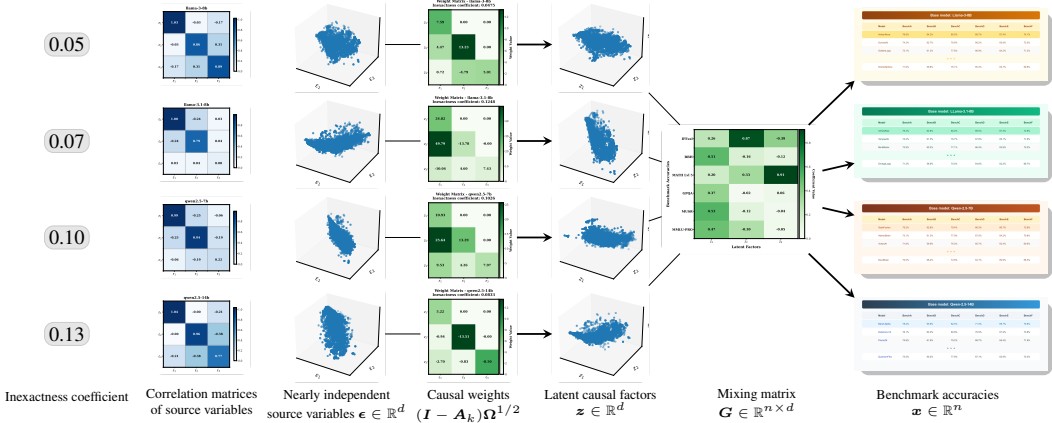

Figure 19: HCA's recovery of the DGP after removing badly fine-tuned models that have average performance lower than the pretrained model.

### G.4 USING OPEN LM LEADERBOARD V1

We also apply our method to analyze the hierarchical structure underlying the six benchmarks used in the old version of open LM leaderboard. We choose the following six base models that are most commonly used there: Mistral-7B, Llama-2-13B, Llama-3-8B, Llama-2-7B, Llama-2-70B and Mixtral-8x7B. Similar to our previous case, we plot the pairwise cosine distance between domains in Figure 22b. We denote these models by $\mathcal{M}_1, \cdots, \mathcal{M}_6$. We observe that except for Llama-2-7B, the principle component subspaces of all remaining domains are pretty close to each other, so that the

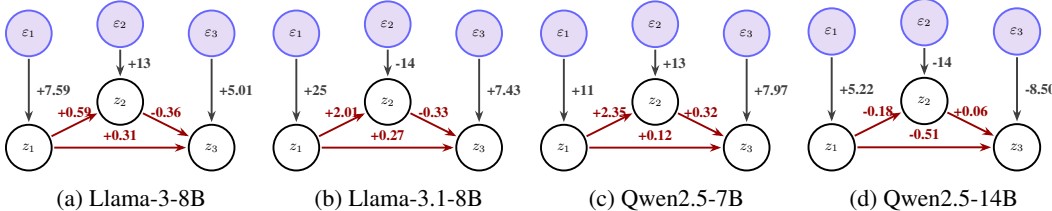

(a) Llama-3-8B  (b) Llama-3.1-8B  (c) Qwen2.5-7B  (d) Qwen2.5-14B

Figure 20: The causal graphs recovered for different models. The numbers represent the weights of each causal edge. For instance, in the Llama-3-8B model, $z_2 = 0.59z_1 + 13\varepsilon_2$ (representing direct influences shown).

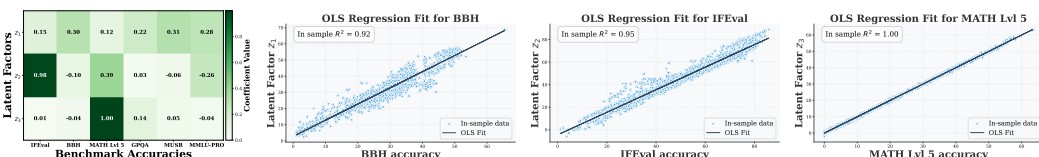

(a) Adjusted unmixing ma- (b) Regressing $z_1$ on BBH. (c) Regressing $z_2$ on IFE- (d) Regressing $z_3$ on trix.  val.  MATH.

Figure 21: The unmixing matrix and the alignment between benchmarks and capabilities via OLS.

invariant domain is $S_{\text{inv}} = \{1, 2, 3, 5, 6\}$. This is quite interesting, since Mixtral-8x7B uses MoE architecture, which is a fundamental difference compared with the other base models.

We then run HCA on all subsets of $S_{\text{inv}}$ of size $\geq 3$ and plot the corresponding MIC in in Figure 22c. We observe that choosing all domains in $S_{\text{inv}}$ would still lead to a small error. The corresponding recovered DGP is presented in Figure 23. We further adjust for the ambiguity as in Section 4, and obtain the causal graphs shown in Figure 24. Finally, the adjusted unmixing matrix and the alignment between latent factors and benchmarks are presented in Figure 25.

From Figure 25, we can see a hierarchical relationship from truthfulness to general reasoning capability, and math reasoning capability. The hierarchical relationship between the latter two is consistent with our findings on the Open LLM Leaderboard v2. By looking at the causal graphs, one can observe that the weight of the edge $z_2 \to z_3$ for the Llama-3-8B domain is much larger that that of the remaining ones, which indicates that models fine-tuned on Llama-3-8B could have more performance gains on math problem solving when fine-tuned to enhance general reasoning.

### G.5 MMLU BY TASK LEADERBOARD

The MMLU benchmark has a total of 57 subtasks, each corresponding to a distinct subject. It therefore makes sense to apply our methodology to these subjects and investigate their latent causal structure. To begin with, we first investigate the correlation between the performance of different tasks in MMLU, which is plotted in Figure 26. We observe that a majority of tasks have highly-correlated performance, although they seemingly focus on unrelated fields. This is likely due to the fact that MMLU primarily contains knowledge-based tasks, and crucially depends on the quality of the training dataset. Larger datasets likely contain more data in all disciplines and can hence lead to improvement on all tasks. In terms of causality, this means that there exists a single "knowledge" node for the MMLU benchmark as a whole.

**Math-related subjects.** We first select subjects that correspond to mathematics, including: MMLU_college_mathematics, MMLU_elementary_mathematics, MMLU_high_school_mathematics. In this setting, we choose the set of base models to be Mistral-7B, Llama-2-7B and Llama-2-70B, which induces a minimal MIC of 0.02. The result of HCA is summarized in Figure 27c. Counter-intuitively, it shows that $z_1$ is close to college math, while $z_2, z_3$ likely represent elementary and high-school math.

**Physics-related subjects.** We conduct a similar analysis for Physics-related subjects. We choose the set of base models to be Mistral-7B, Mistral-8x7B, Llama-2-13B, Llama-2-70B, which induce a minimal MIC of 0.05 among all domain subsets with size $\geq 4$. The result of HCA is summarized in

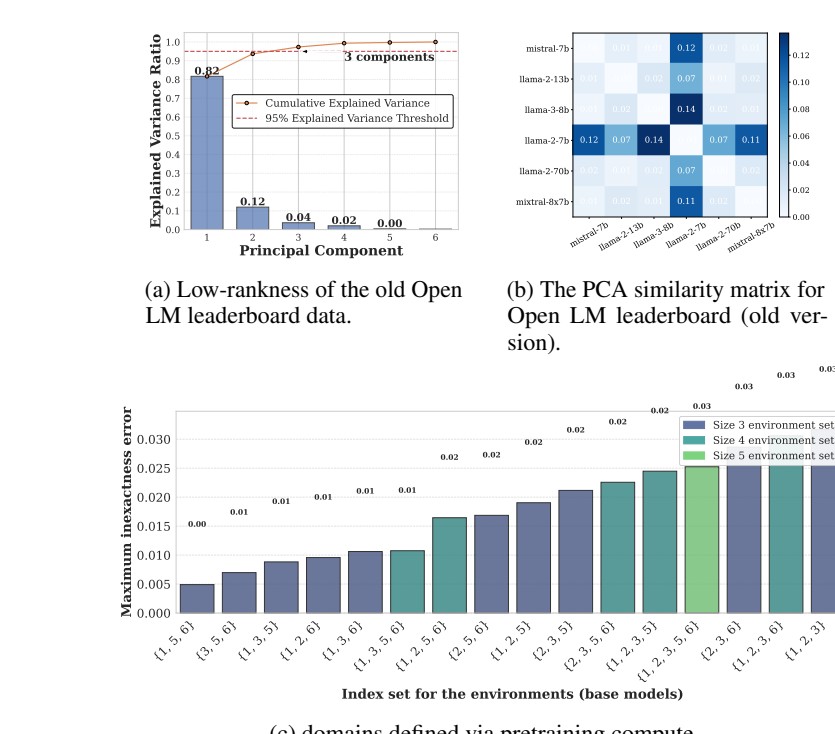

(a) Low-rankness of the old Open LM leaderboard data.

(b) The PCA similarity matrix for Open LM leaderboard (old version).

(c) domains defined via pretraining compute.

Figure 22: figures for our analysis of Open LLM leaderboard v1.

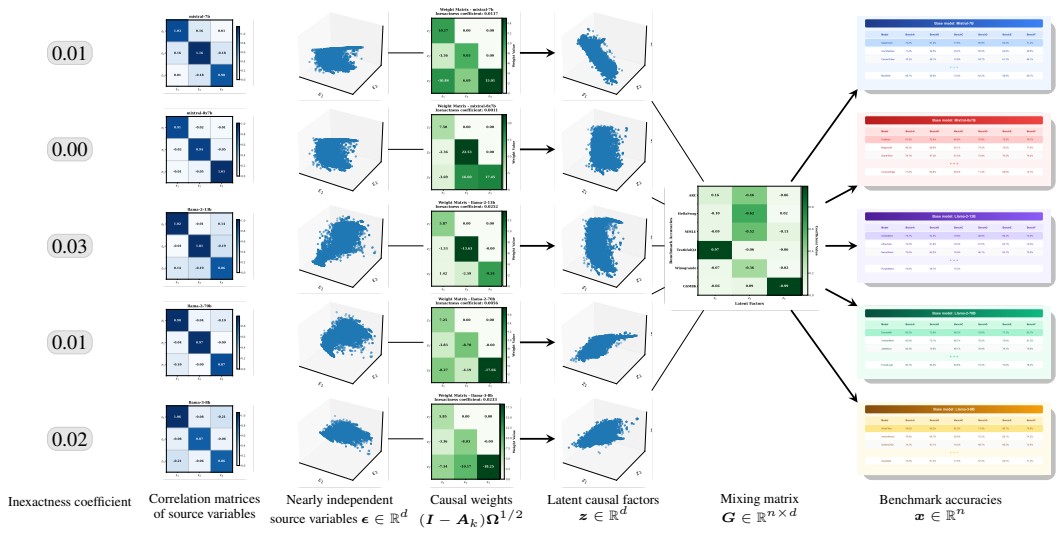

Figure 23: Results for applying our method to open LM leaderboard v1.

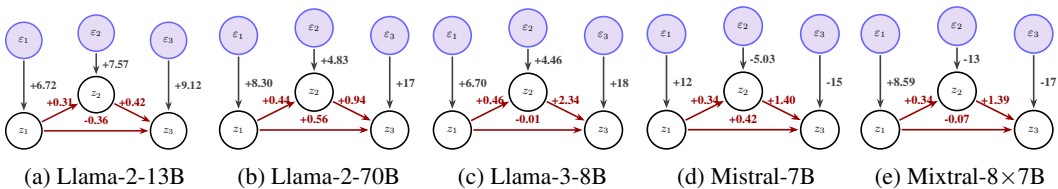

(a) Llama-2-13B  (b) Llama-2-70B  (c) Llama-3-8B  (d) Mistral-7B  (e) Mixtral-8×7B

Figure 24: The causal graphs recovered for different models. The numbers represent the weights of each causal edge. For instance, in the Llama-2-13B model, $z_2 = 0.31 z_1 + 7.57\varepsilon_2$ (representing direct influences shown).

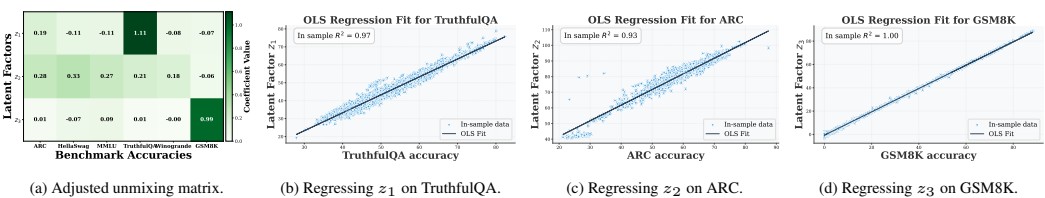

(a) Adjusted unmixing matrix.    (b) Regressing $z_1$ on TruthfulQA.    (c) Regressing $z_2$ on ARC.    (d) Regressing $z_3$ on GSM8K.

Figure 25: The unmixing matrix and the alignment between benchmarks and capabilities via OLS.

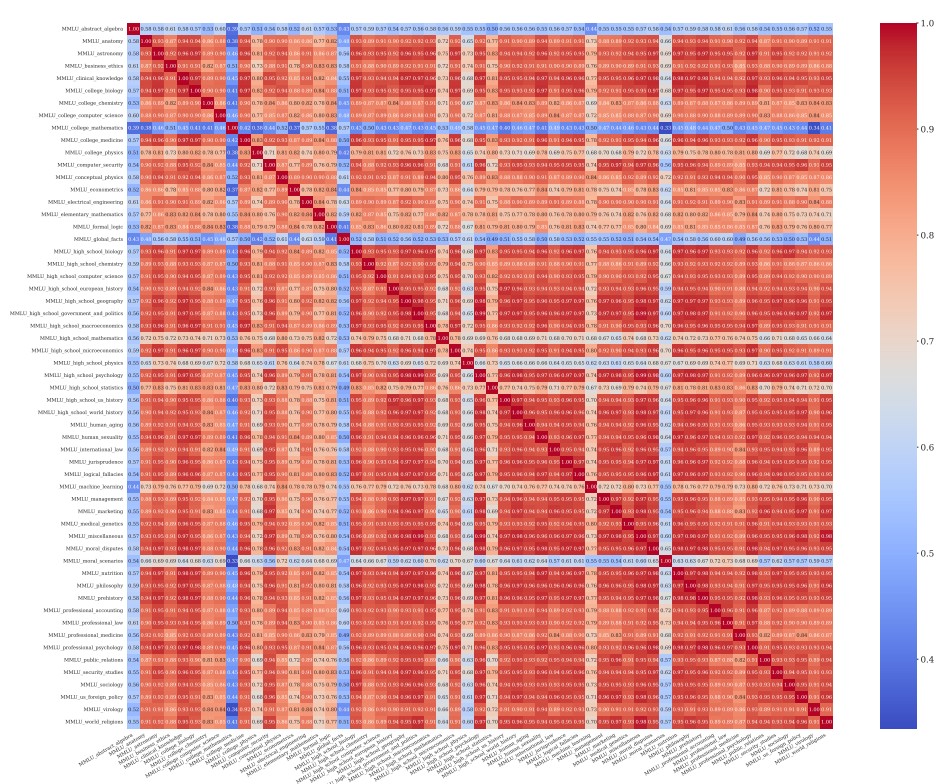

Figure 26: Correlation matrix for the tasks in the MMLU benchmark.

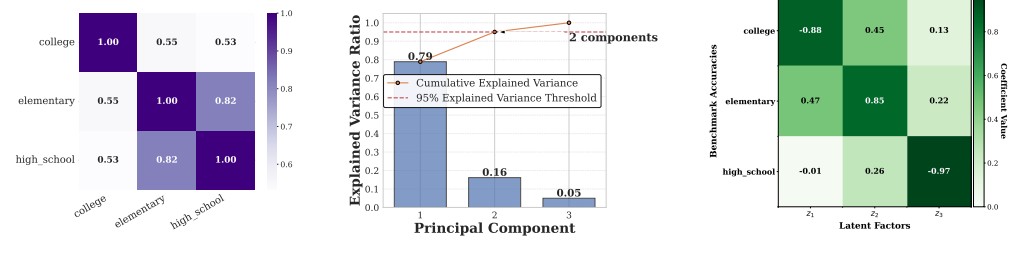

(a) Correlations of the performance of math-related subtasks.

(b) Low-rank structure of the subtasks performance data.

(c) The learned mixing matrix.

Figure 27: HCA analysis of the MMLU by Task Leaderboard data of math-related subjects.

Figure 28c. We can see that $z_1$ is conceptual physics while $z_2$ and $z_3$ are both linear cominations of high-school and college physics.

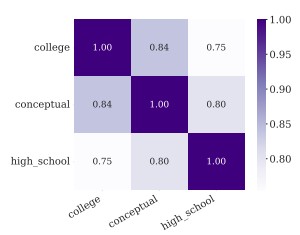
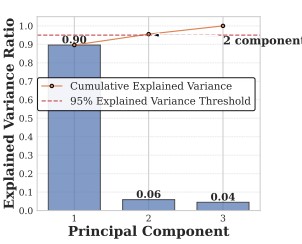
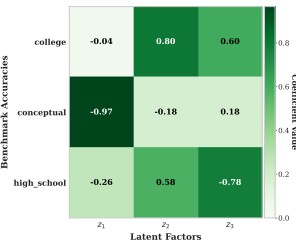

(a) Correlations of the performance of physics-related subtasks.

(b) Low-rank structure of the subtasks performance data.

(c) The learned mixing matrix.

Figure 28: HCA analysis of the MMLU by Task Leaderboard data of physics-related subjects.

**Cross-subject domains.** It would also be interesting to explore how different subjects are related. We choose MMLU_college_mathematics, MMLU_college_physics and MMLU_college_electrical_engineering and run HCA on these subjects. We found the a hierarchical relationship exists in the order of math, electrical engineering and physics.

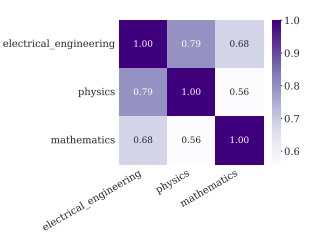
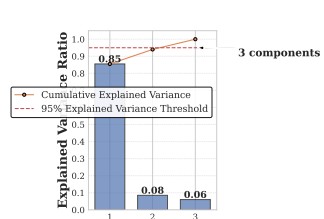
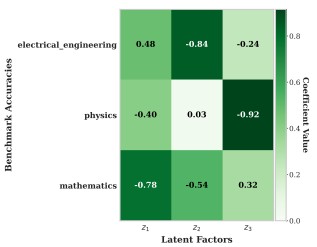

(a) Correlations of the performance of tasks under different subjects.

(b) Low-rank structure of the subtasks performance data.

(c) The learned mixing matrix.

Figure 29: HCA analysis of the MMLU by Task Leaderboard data of three different subjects

## H  SENSITIVITY ANALYSIS OF CAUSAL GRAPH RECOVERY

While very few works in the literature study finite sample guarantees for causal representation learning (Acartürk et al., 2024), the accuracy of the recovered causal graph is crucial for downstream scientific studies. In this section, we conduct a sensitivity analysis of the causal model we discover, and discuss its implications. Specifically,

- The causal model *could* be sensitive to the choice of base models. We discuss possible reasons for this.
- We find that the causal link between instruction following and math reasoning is stable.

**Adding Two Base Models.** We add two more base models, Qwen2.5-3B and Llama-3.2-3B, into our causal analysis. From Figure 10 we can see that these two domains have roughly the same PC subspace as the four domains we previously used in the main paper. Therefore, Hypothesis 1 would not be violated. That said, one caveat is that these two domains only include $65$ and $94$ models respectively, while the four previously used domains all include more than $150$ models.

We run the same algorithms on these six domains, and the results are summarized in Figure 31 and Figure 32. One can see that the pattern of the unmixing matrix, as shown in Figure 31a, is the same as Figure 7a in $z_2$ ($\approx$ IFEval) and $z_3$ ($\approx$ MATH Lvl 5) but different in $z_1$. In particular, rather than

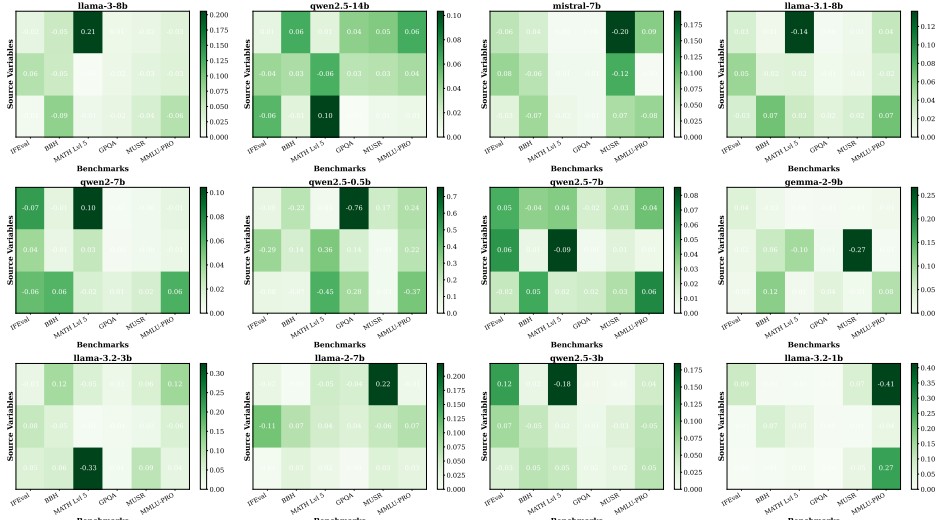

Figure 30: The unmixing matrices of ICA for individual domains.

being closely aligned with general reasoning benchmarks like BBH and MMLU-Pro, it represents some sort of tradeoff between general reasoning and specialized capabilities.

Recall that given infinite samples, we show that the causal model is identifiable up to mixtures with ancestors, wich does not include the case of Figure 31a. Recall that for picking $z_1$, the key idea was to find a combination of rows in $M_k$ that are colinear. In the finite sample regime and/or our causal assumptions are not exact, the error induced by this step could be hard to control.

In Figure 30 we present the ICA unmixing matrices $M_k$. We find that among the domains chosen to conduct the causal analysis, there are two notable patterns:

1. There is a row that is approximately a weighted combination of BBH and MMLU-Pro, *e.g.* third row in the matrix of Llama-3-8B. Recall that these two benchmarks are highly correlated, so this is close to the general reasoning capability discovered in Figure 7a as $z_1$.

2. There is another row that has positive weights on BBH and MMLU-Pro and negative weights on IFEval and MATH (or vice versa), *e.g.* the second row of Llama-3-8B. This row is aligned with the $z_1$ that we discover here in Figure 31a.

These observations indicate that one possible cause for the sensitivity of causal graphs is the environment non-degeneracy assumption (Assumption 2). even when the data comes from an exact causal graph, in the finite sample case, when there exists two $i$'s such that $\text{span}\langle (B_k)_i, k \in [K] \rangle$ is approximately rank-1, the corresponding causal factor can be hard to identify.

Note that for base models that are not used in our causal analysis, namely Mistral-7B, Qwen2-7B, Qwen2.5-0.5B, Gemma-2-9B, Llama-2-7B and Llama-3.2-1B, the unmixing matrices do not demonstrate these patterns. This is a paritcularly interesting observation that we left for future studies, since it likely reveals different latent knowledges learned by different base models.

Since algorithm might pick either one of the above as $z_1$, and the induced errors are comparable, it is hard to argure which one makes more sense. Nonetheless, we find that the subspace spanned by the first two rows of Figure 7a and Figure 31a are roughly the same, and IFEval approximately lies in this subspace. Moreover, $z_3$ always represents math reasoning. So, the causal effect from instruction following to math reasoning is stable across different choices of domains. Moreover, viewing the weight of the causal edge $z_2 \rightarrow z_3$ as the causal effect of fine-tuning on instruction following, we can see that this effect is generally much higher for Qwen models compared with Llama models, a phenomenon that we also observe in Figure 6. We also notice that the estimated effect is roughly the same for Qwen models in these two approaches, while for Llama models, the weights are smaller here compared with the ones in Figure 6. Overall, further investigating the effect of finite-sample error and violations of causal assumptions is an important direction for future research.

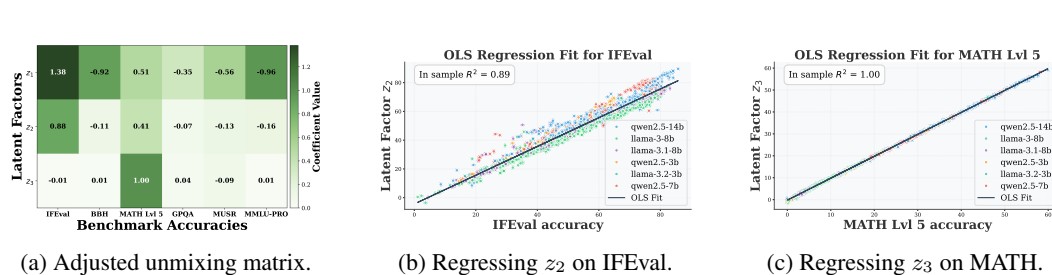

(a) Adjusted unmixing matrix.  (b) Regressing $z_2$ on IFEval.  (c) Regressing $z_3$ on MATH.

Figure 31: The unmixing matrix and the alignment between benchmarks and capabilities via OLS.

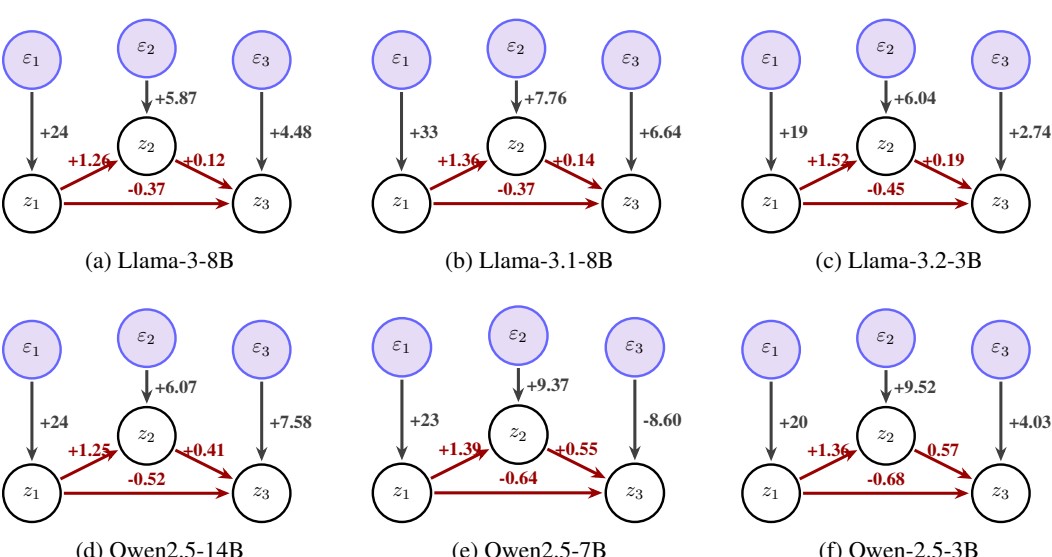

Figure 32: Causal graphs recovered from each domain.

