# OpenReview forum: "Discovering Hierarchical Latent Capabilities of Language Models via Causal Representation Learning"
_ICLR.cc/2026/Conference — Submitted to ICLR 2026_

### Official Review · Reviewer_ay8L · 2025-10-22

**Soundness:** 2
**Presentation:** 3
**Contribution:** 3
**Rating:** 6
**Confidence:** 2

**Summary:**

The authors introduce an $\alpha$-inexact structural causal model (SCM) for LLM benchmark scores. A novel identification procedure that improves upon sensitivity to the inexactness, as compared to prior work, is presented. Utilizing their model and algorithm, the authors produce three latent LLM ``skill" factors and their causal relationship.

**Strengths:**

* The SCM model allows the authors to uncover an interesting causal relationship between generalist, instruction following, and mathematical skills in LLMs
* The HCA algorithm comes with theoretical guarantees in the exact and inexact SCM settings.

**Weaknesses:**

* Compared to prior work (Polo et al.) the model is fit and studied on a relatively smaller number of benchmarks/models. The authors could consider utilizing LLM leaderboard v1 and v2.
* Besides the exactness assumptionsThe SCM is identical to that proposed in Jin & Syrgkanis; Zhang et al. Furthermore, I am unable to quantify how HCA is an improvement over the methods in these works. Proposition . shows that if ICA is exact then HCA recovers an $\alpha$-exact SCM. The authors state that prior methods are sensitive to (in)exactness but never explain further (in particular I would be curious if the exact ICA assumption affects this at all).
* I am not sure that the discussion in section 2 is a totally correct assessment of prior work on latent LLM skills. For example, if I understand (Polo et al.) the key assumption is $x \approx Gz(s,t)$, with s, t model size and token count. To me this implicitly allows the latent space to vary based on the model family, which is the main argument for the model the authors introduce.

**Small writing issues**

* Line 320: ''varisbles''
* Line 265 $z^{(k, i)}$ appears on both sides of the equation, im not sure how this should read

**Questions:**

* How do the results extend to the benchmarks and models present in llm leaderboard v1/v2?
* Can the authors provide a more thorough comparison between HCA and prior algorithms in the $\alpha$-inexact setting?

---

> ### Author Response · Authors · 2025-11-14
>
> We thank the reviewer for the positive feedback and for raising insightful questions and concerns. Below is our response to them.
>
> **Weakness #1: Compared to prior work (Polo et al.) the model is fit and studied on a relatively smaller number of benchmarks/models. The authors could consider utilizing LLM leaderboard v1 and v2.**
>
> If our understanding is correct, same as Polo et al. we also use Open LLM Leaderboard v1/v2 in our analysis. The main part of the paper considers v2, while analysis for v1 can be found in Appendix G.4 due to space limitations. There are indeed differences in the model families that we use versus Polo et al. This is because we choose a different way to model the capabilities. Specifically, we are interested in discovering causal links between capabilities, and the hypothesis that “each model family has the same latent capability structure” is not true in general. In our work, we select model families that share the same latent capability structure (which is verifiable from data) and do not consider other model families. In short, our characterization of capabilities is more fine-grained compared with prior analysis, so is only applicable to a subset of model families. Our final result covers four base models and more than 1000 models on the Open LLM Leaderboard.
>
> **Weakness #2: Besides the exactness assumptionsThe SCM is identical to that proposed in Jin & Syrgkanis; Zhang et al. Furthermore, I am unable to quantify how HCA is an improvement over the methods in these works. Proposition . shows that if ICA is exact then HCA recovers an -exact SCM. The authors state that prior methods are sensitive to (in)exactness but never explain further (in particular I would be curious if the exact ICA assumption affects this at all).**
>
> The reviewer raises two questions here: first, how to understand the relationship between HCA and prior algorithms. Second, the effect of assumptions about the ICA step.
>
> Answer to the first question: yes, HCA shares the same key idea as the LiNGCReL algorithm proposed in Jin and Syrgkanis’ work. We do make some changes though, and details can be found in Appendix C.1. Roughly speaking, due to inexactness of the causal assumptions, we wish to recover a causal model that is robust to small model misspecification errors. We also relax the assumption that the noise distributions are the same across domains. That said, at a high level, these two algorithms have no big difference.
>
> Compared to Zhang’s work, we study a linear causal model rather than a nonlinear one. Although linearity is a restrictive assumption, we develop a diagnosis of whether the given dataset actually satisfies the assumption. Moreover, methods for nonlinear causal models often involve complicated optimization steps and is not an ideal choice in our setting, where evaluation data is limited.
>
> Answer to the second question: exactness of the ICA step is required, since HCA uses ICA as a first step to recover the product of weight matrix and mixing matrix, so theoretically, exact identification of HCA requires that ICA is also exact.
>
> **Weakness #3: I am not sure that the discussion in section 2 is a totally correct assessment of prior work on latent LLM skills. For example, if I understand (Polo et al.) implicitly allows the latent space to vary based on the model family, which is the main argument for the model the authors introduce.**
>
> That’s a very good point! There is a subtle but crucial difference between defining the domains based on model/token size versus model families. Two base models with similar model and token sizes could have very different downstream performance. A simple example is that Llama-3-8B with 15T training tokens is roughly comparable with Qwen2.5-7B with 18T training tokens, but is much worse on mathematical reasoning (see e.g. Fig. (12) in our paper). The root cause might be that Qwen models use much more math-related data during pretraining, a feature that is not captured by model size or token size. Therefore, we view splitting via base model family as fundamentally different from splitting via model/token size.

---

> > ### Author Response · Authors · 2025-11-14
> >
> > **Weakness #4: small writing issues**
> >
> > We thank the reviewer for pointing these out! It is true that in line 265, $z^{(k,i)}$ appears on both sides. But since $A_k$ is a matrix that respects the underlying causal DAG, each $z^{(k,i)}$ in fact only depends on its “causal parents” in the DAG. Hence it is well-defined. We will make this more explicit in a revised version.
> >
> > **Question #1: How do the results extend to the benchmarks and models present in llm leaderboard v1/v2?**
> >
> > Actually our main results are already established for the open LLM leaderboards v1/v2 (results for v1 can be found in appendix). We would like to thank the reviewer for letting us realize the possible confusion here,and we will further clarify in an updated version.
> >
> > **Question #2: Can the authors provide a more thorough comparison between HCA and prior algorithms in the $\alpha$-inexact setting?**
> >
> > HCA is essentially the only known algorithm in this setting that can guarantee identifiability. The $\alpha$-inexactness for causal graphs also appears to be novel. We would like to emphasize that this work does not try to improve upon any existing results or benchmarks. Rather it introduces a new perspective of causally related capability that previous works do not explore. All algorithms and theories are carefully chosen and designed to match this new perspective.
> >
> > We thank the reviewer again for the effort in the reviewing process, and we look forward to hearing back from the reviewer in case there are any additional questions.

---

### Official Review · Reviewer_x9Ag · 2025-10-26

**Soundness:** 1
**Presentation:** 2
**Contribution:** 2
**Rating:** 4
**Confidence:** 3

**Summary:**

This paper introduces Hierarchical Component Analysis (HCA), a novel framework inspired by causal representation learning, designed to uncover the hierarchical causal relationships among latent capabilities of LLMs using solely publicly available observational benchmark data. Addressing the challenges of confounding effects and the high cost of controlled studies in LLM evaluation, the authors model observed performance as a linear transformation of a few latent capabilities, crucially assuming these capabilities form a DAG after controlling for the base model as a confounder. The study identifies a consistent three-node linear causal hierarchy: general problem-solving influences instruction-following, which in turn influences mathematical reasoning.

**Strengths:**

1. The paper pioneers the application of causal representation learning using only observational data to understand LLM capabilities, moving beyond correlational scaling laws or factor analysis.
2. The discovered hierarchy (general problem-solving → instruction-following → math reasoning) provides a potentially practical "causal roadmap" for prioritizing post-training efforts, suggesting interventions on parent capabilities can benefit child capabilities.

**Weaknesses:**

1. The method relies on strong assumptions like the linearity of capability-performance mapping, the DAG structure of capabilities, and non-Gaussian noise for ICA, which might not fully hold in reality . The impact of violations (e.g., non-linearity) is not deeply explored.

2. Concerns were raised about the discovered graph's robustness to the choice of base models and benchmarks. While Appendix H provides some sensitivity analysis, the filtering step and potential finite-sample errors impacting ICA/HCA need consideration. The complexity of Algorithm 2's permutation search could be an issue for higher dimensions, though feasible for d=3.

3. In fact, I was one of the reviewers of this paper submitting to Neurips. At that time, the chair and other reviewers were more concerned about issues around assumption realism, experimental scope/evaluation breadth, and practical utility. However, when I reread the author's submission this time, I did not find that the author had added justification to address these issues, which led to my support for this paper declining.

**Questions:**

1. How sensitive is the recovered hierarchy ($z_1 \rightarrow z_2 \rightarrow z_3$) to potential non-linear relationships between capabilities or non-linear mappings to benchmark scores? How is the "inexactness" (quantified by MIC) distributed across different causal links or domains?

2. How likely is this specific 3-node hierarchy to generalize to other base model families (beyond Llama-3/Qwen2.5), different model scales, or architectures like MoEs ? Does the exclusion of models with different PC subspaces (Fig 2c) limit the generality of the findings?

3. Beyond correlation with benchmarks (BBH, IFEval, MATH), what is the deeper semantic meaning of $z_1, z_2, z_3$? Is $z_1$ purely general reasoning or confounded with knowledge? How distinct are these factors, especially given the ambiguities allowed by the identification theory (Theorem 1)?

---

> ### Author Response · Authors · 2025-11-14
>
> We thank the reviewer for the insightful feedback. As a starting point, we would like to share our view of the contributions of this paper in one sentence:
>
> *This paper introduces a novel way to generate hypotheses regarding the causal relationship between LLM capabilities, based solely on observational data.*
>
> We believe it helps to clarify that this paper is *not* trying to establish any *universal causal rule* for ‘the world of capabilities’. Rather, it only tells us what causal links likely exist, which can then be verified via fine-tuning experiments. We build the theoretical and methodological foundations that allow such hypotheses to be generated solely from observational leaderboard data.
>
> Below, we provide our response to the weaknesses that the reviewer points out and the questions that the reviewer has.
>
> **Weakness #1: The method relies on strong assumptions like the linearity of capability-performance mapping, the DAG structure of capabilities, and non-Gaussian noise for ICA, which might not fully hold in reality . The impact of violations (e.g., non-linearity) is not deeply explored.**
>
> This is a valid concern and, in fact, a common concern from reviewers. Let us analyze these assumptions one by one.
>
> Linearity of capability-performance mapping: this is a common assumption in previous works on ‘observational scaling laws’ [1,2,3]. These works find that the model-performance matrix is often low rank, and motivates the hypothesis that the benchmark performances are driven by linear combinations of a few capabilities. We verify the same property in our work.
>
> It is a little bit ambiguous to argue whether the linearity assumption holds in practice, because, as the reviewer has noticed in the review, the ‘capabilities’ are fully unknown apriori. Previous works [1,2,3] use factor models like PCA and define capabilities to be the principal component. In principle, one can choose any linear combinations of benchmarks and call them ‘capabilities’. We are thus motivated to investigate ‘what type of capability representations can actually be useful?’ This motivates us to introduce the next hypothesis: DAG structure among capabilities.
>
> So, is DAG structure a reasonable assumption? As we just mentioned, we introduce this assumption because we want to discover causal relationships among capabilities, which can be useful for fine-tuning. However, two concerns may arise: 1. whether additional ambiguities would arise here, i.e. there are multiple ways to specify the DAG. 2. We assume linear DAG, but it can actually be nonlinear.
>
> We address both concerns in our work. For the first one, we build identifiability theory to characterize the ambiguity. It turns out that only one ambiguity could arise, namely mixtures of each capability $z_i$ with its causal parents. For instance if $z_1\to z_2$, one cannot distinguish between $z_2$ and $z_2+\alpha z_1$. It is worth noticing that the causal interpretation remains the same, only the weight of this edge could change when a different $\alpha$ is chosen.
>
> For the second one, we introduce MIC to quantify the error induced by fitting a linear DAG. Admittedly, the linearity assumption does not always hold, and in this paper we do not deal with the non-linear case. However, we do know whether the assumed linear model is a good-fit for the observed data. In this paper, we show that this is indeed the case, at least across the model families we study.
>
>
> The non-Gaussian noise assumption is not a strong one – indeed, the noise should always be uniformly bounded since all benchmark performances should lie in [0,1]. Assuming non-Gaussianity allows us to establish the aforementioned identifiability theory.
>
> We hope that this addresses the reviewer’s concerns about the assumptions. to summarize:
> The assumptions are motivated by the need of fine-tuning.
> The assumptions are not guaranteed to hold but we have a good understanding of whether (more precisely, to what extent) it holds for any given dataset.
>
> [1]. Ruan, Yangjun, Chris J. Maddison, and Tatsunori B. Hashimoto. "Observational scaling laws and the predictability of langauge model performance." NeurIPS 2024.
>
> [2]. Ren, Richard, et al. "Safetywashing: Do AI Safety Benchmarks Actually Measure Safety Progress?." NeurIPS 2024.
>
> [3]. Polo, Felipe Maia, et al. "Sloth: scaling laws for LLM skills to predict multi-benchmark performance across families." NeurIPS 2025.

---

> > ### Author Response · Authors · 2025-11-14
> >
> > **Weakness #2: Concerns were raised about the discovered graph's robustness to the choice of base models and benchmarks. While Appendix H provides some sensitivity analysis, the filtering step and potential finite-sample errors impacting ICA/HCA need consideration. The complexity of Algorithm 2's permutation search could be an issue for higher dimensions, though feasible for d=3.**
> >
> > Good question!
> >
> > Regarding the choice of base models: the filtering step (that removes some base models from our study) is not ‘artificial’. Rather, as we pointed out in the response to the previous weakness point, we do have a good diagnosis of whether the dataset (corresponding to an arbitrary set of base models) can indeed fit the linear causal model that we specify. Some base models are removed because they don’t. As illustrated in Figure 2(c), base model families like Qwen2.5-0.5B induce a different principal component subspace.
> >
> > Regarding the choice of benchmarks: we use the six benchmarks from the open LLM leaderboard because we do not have any other benchmarks available. In the appendix, we also experiment on the old version of the leaderboard (also has six benchmarks, but the evaluated models are different) and MMLU-by-task leaderboard. While our causal graph does not cover all capabilities, it does provide insights into the hierarchy of capabilities that are ‘present’ in the available benchmarks. The algorithm’s complexity is a real issue and we would expect that the current algorithm does not work for larger d’s. Generally speaking, the complexity of causal representation learning algorithms is an underexplored theoretical problem, with only partial results [4] that do not directly apply to our setting.
> >
> > Lastly, the finite-sample error of HCA/ICA indeed needs to be considered. Our current algorithm (HCA) could have potential errors from three sources: 1. model mis-specification of ICA; 2. finite-sample error for ICA and 3. approximation error of the linear causal model. We quantify the last error source by MIC. The second error source is well studied in the literature [5] and we intentionally choose to work with leaderboards that contain a large number of models.  We are unaware of a universal solution or error metric for the first error source though. However, as mentioned at the beginning of the rebuttal, we are not merely running an off-the-shelf algorithm to reach a conclusion. Eventually, our hypothesis is evaluated via actual fine-tuning experiments.
> >
> > [4]. Acartürk, Emre, et al. "Sample complexity of interventional causal representation learning." Advances in Neural Information Processing Systems 37 (2024): 39350-39385.
> >
> > [5]. Auddy, Arnab, and Ming Yuan. "Large-dimensional independent component analysis: Statistical optimality and computational tractability." The Annals of Statistics 53.2 (2025): 477-505.
> >
> >
> > **Weakness #3: In fact, I was one of the reviewers of this paper submitting to Neurips. At that time, the chair and other reviewers were more concerned about issues around assumption realism, experimental scope/evaluation breadth, and practical utility. However, when I reread the author's submission this time, I did not find that the author had added justification to address these issues, which led to my support for this paper declining.**
> >
> > We thank the reviewer for sharing this piece of information. Although we don’t really know what has been discussed between the reviewers and AC, we do get the same message when reading the review’s and the AC’s final remarks. We did not make significant changes to this ICLR submission, because we felt the main issue was that the setting, contributions  and comparison with existing works were not conveyed clearly in the previous version. We mainly improve some of the writing to better position our work in the literature. We answered a very similar set of questions raised by the reviewers during NeurIPS rebuttal, but it was unclear whether our responses indeed addressed all the concerns from the reviewers and the AC. We greatly appreciate the reviewer’s effort in reviewing the same work twice, and love to hear any extra insights in terms of what can be further improved.
> >
> > Additionally, we feel that it might be helpful to share some of our final thoughts after the NeurIPS submission. The main novelty of this work lies in causal modelling of LLM capabilities and leveraging causal representation learning (CRL) techniques to recover the model. Despite the well-developed theories in this field, to the best of our knowledge, no existing works have explored along this direction, so it is not directly comparable to any other papers. Generally speaking, it is difficult to anticipate that a rigorous statistical theory can perfectly match an application on real data. ‘All models are wrong, but some are useful.’ Our work is primarily driven by the usefulness of our new perspective, and at the same time, we tried our best to provide justifications and validation for our assumptions.

---

> > > ### Author Response · Authors · 2025-11-14
> > >
> > > **Question #1: How sensitive is the recovered hierarchy to potential non-linear relationships between capabilities or non-linear mappings to benchmark scores? How is the "inexactness" (quantified by MIC) distributed across different causal links or domains?**
> > >
> > > As we explained in the response to weakness #1, there are two cases:
> > >
> > > - If a linear model is sufficient to capture the relationships between capabilities and mapping to benchmark scores, then our current analysis is valid.
> > >
> > > - If it is not, we can detect it by checking the pairwise distances between principal component spaces of different model families (figure 2(c)) and the MIC.
> > >
> > > Hence, it is in fact possible to determine whether we should use a linear model or not. There are also some identifiability results for nonlinear models [6], but because that is a more general setting, one should expect to recover a linear model if the ground-truth is actually linear. An interesting future direction is to address the second case, but currently there is no end-to-end identifiability guarantee for nonlinear models due to optimization challenges. Finally, the MIC is the maximal error across all domains. Within each domain, there could be a tradeoff in terms of the errors of each causal link, and in line 993 we explicitly choose the ‘best alignment’ for each domain.
> > >
> > > [6]. Zhang, Kun, et al. "Causal Representation Learning from Multiple Distributions: A General Setting." International Conference on Machine Learning. PMLR, 2024.
> > >
> > >
> > > **Question #2: How likely is this specific 3-node hierarchy to generalize to other base model families (beyond Llama-3/Qwen2.5), different model scales, or architectures like MoEs ? Does the exclusion of models with different PC subspaces (Fig 2c) limit the generality of the findings?**
> > >
> > > Since our analyses are primarily based on Open LLM Leaderboard, we only consider several most frequently used base models to guarantee that we have enough samples for each one. Among them, some base models are excluded as the reviewer noticed. While this indeed limits the generality of the findings, we view this as a *necessary step* to avoid drawing misleading conclusions for model families where the capability hierarchy doesn’t really apply. It is possible that while the whole graph does not apply to other model families, some causal links (e.g. $z_2 \to z_3$) remains valid, but it requires novel algorithmic tools and we leave it to future works.
> > >
> > > Within each model family we also distinguish between different model scales (e.g., Qwen2.5-0.5/7/14B). We did not consider the MoE architecture since the corresponding base models (mainly mixtral) are too rare in the leaderboard. However, we do include mixtral-8x7b in the analysis based on the old leaderboard; see Page 31.
> > >
> > >
> > > **Question #3:  Beyond correlation with benchmarks (BBH, IFEval, MATH), what is the deeper semantic meaning of $z_1$ ? Is it purely general reasoning or confounded with knowledge? How distinct are these factors, especially given the ambiguities allowed by the identification theory (Theorem 1)?**
> > >
> > > That’s a great question. We find it hard to argue about the “semantic” meaning of a latent factor and the best thing that we can do is to examine the correlation with benchmarks. This is a reasonable surrogate, since the question we ultimately care about is, “what does it mean to intervene on $z_1$?” This should correspond to a set of observed variables (benchmark performances) and can be approximated by the performances after fine-tuning on a relevant type of task.
> > >
> > > As we explain in the response to Weakness #1, the identification ambiguity changes the precise meaning of each causal factor but does not change the overall causal structure. For example, “A increased by 1 => B increased by 0.5” is equivalent to saying that “A increased by 1 => A+B increased by 1.5”. This is precisely (and the only) ambiguity as characterized in Theorem 1.
> > >
> > > We thank the reviewer again for the effort in the reviewing process, and we look forward to hearing back from the reviewer in case there are any additional questions.

---

### Official Review · Reviewer_x3s1 · 2025-10-29

**Soundness:** 2
**Presentation:** 2
**Contribution:** 2
**Rating:** 4
**Confidence:** 4

**Summary:**

The paper introduces a novel causal representation learning (CRL) framework, Hierarchical Component Analysis (HCA), to discover the hierarchical structure of latent capabilities in LLMs using only observational benchmark data from the Open LLM Leaderboard. By controlling for the base model as a common confounder and analyzing latent factors (Hypothesis 1) through a linear Structural Causal Model (SCM) (Hypothesis 2), HCA recovers a three-node hierarchy: General Problem-Solving ($z_1$) $\rightarrow$ Instruction-Following ($z_2$) $\rightarrow$ Advanced Mathematical Reasoning ($z_3$). This validated causal roadmap offers practitioners clear, actionable insights for strategic post-training interventions.

**Strengths:**

1. Causal Discovery of Hierarchical Structure: The HCA method moves beyond correlation-based analysis (like PCA) to establish a causally-directed hierarchy among capabilities, providing a validated roadmap for improving LLMs.
2. Robustness by Controlling Confounding: The analysis rigorously accounts for performance heterogeneity across different base models, ensuring the discovered invariant causal structure is robust and not merely a result of pre-training confounding effects.
3. Actionable Insights for Optimization: The hierarchy provides practical guidance, such as demonstrating that enhancing the intermediate Instruction-Following capability ($z_2$) is an effective precursor for boosting the specialized Mathematical Reasoning skill ($z_3$).

**Weaknesses:**

1. The analysis is constrained by the six benchmarks available, resulting in only three coarse-grained capabilities. A richer set of benchmarks might reveal a more complex hierarchy .
2. While SFT experiments support the $z_2 \rightarrow z_3$ link, the broader causal claims rely partly on observational correlations and interpretation. The ATE analysis acknowledges unverifiable ignorability assumptions. There could be the potential circularity in naming factors based on correlated benchmarks and then using those benchmarks for validation.

**Questions:**

1. How exactly would practitioners map real-world fine-tuning strategies (e.g., specific datasets, RLHF techniques) onto interventions on the latent factors $z_1, z_2, z_3$ beyond the examples given (IFEval SFT for $z_2$)? How can one target $z_3$ (math) more directly without catastrophic forgetting, as noted in the paper?
2. What is the justification for the specific definition of MIC in Definition 2? Could alternative CRL algorithms designed for multi-node interventions  yield different results? How sensitive is the result to the specifics of the ICA implementation or the permutation alignment process in HCA (Algorithm 2) ?
3. The paper links $z_1$ to pre-training compute (Fig 8). How exactly does pre-training compute act as a confounder influencing all capabilities, and how does HCA effectively control for this beyond simple regression adjustment?

---

> ### Author Response · Authors · 2025-11-14
>
> We thank the reviewer for the insightful feedback and for raising constructive questions and concerns. Below is our response:
>
> **Weakness #1: The analysis is constrained by the six benchmarks available, resulting in only three coarse-grained capabilities. A richer set of benchmarks might reveal a more complex hierarchy.**
>
> This is true – our analysis only uses six benchmarks since the Open LLM Leaderboard only contains six. However, we do apply our methodology to other leaderboards such as the old Open LLM Leaderboard and MMLU by task Leaderboard (albeit with a different set of models being evaluated), demonstrating the generality of our approach.
>
> **Weakness #2: While SFT experiments support the $z_2 \to z_3$  link, the broader causal claims rely partly on observational correlations and interpretation. The ATE analysis acknowledges unverifiable ignorability assumptions. There could be the potential circularity in naming factors based on correlated benchmarks and then using those benchmarks for validation.**
>
> The reviewer raises two questions here: 1. How can we understand the correlation-based interpretation of each capability and 2. unverifiable assumptions for ATE estimation in our setting.
>
> For the first question, as the reviewer correctly points out, our interpretation of each latent causal factor is based on correlation, which the reviewer suspects can lead to circularity issues. We use this approach because it is hard to assign semantic meaning directly to the learned latent variables, so comparing them with existing benchmarks could give us a proxy. From a causal perspective, what our analysis actually reveals is “what would happen under interventions to the latent factors” (Pearl’s causal hierarchy). High correlation between a latent factor and a benchmark allows us to view fine-tuning on particular tasks as a proxy to the true, single-factor interventions.
>
> It is true that this proxy may not always be accurate enough. However, our causal analysis mainly plays the role of a *hypothesis generator* for the causal links between capabilities, rather than an end-to-end guidance to fine-tuning practice. Our hypothesis eventually needs to be verified through experiments. The main advantage of our approach is that hypotheses can be generated by only using observational evaluation data rather than controlled experiments.
>
> For the second question, we guess that the reviewer is referring to Remark 3 in the paper. As the reviewer points out, naive ATE arguments could be problematic since the conditional ignorability assumption is not justified. That remark only serves as a side evidence to provide a more intuitive understanding of the post-training effect across different benchmarks. It is important to note that *our causal discovery method in the main paper is fundamentally different from the ATE analysis in Remark 3*. The ATE analysis is correlation-based. One can view either one of $z_2$ and $z_3$ as “treatment” and the other as “outcome”. Due to positive correlation, there would always be a positive effect. But this tells us nothing about the hierarchy, i.e., which one is indeed the cause.
>
> This type of challenge is ubiquitous in causal analysis. An important observation (used in our approach) to break symmetry is that the cause would have a smaller “degree of freedom” than the effect, and that is in fact the main idea behind our identification arguments.

---

> > ### Author Response · Authors · 2025-11-14
> >
> > **Question #1: How exactly would practitioners map real-world fine-tuning strategies (e.g., specific datasets, RLHF techniques) onto interventions on the latent factors beyond the examples given (IFEval SFT for $z_2$)? How can one target  (math) more directly without catastrophic forgetting, as noted in the paper?**
> >
> > These are very good questions!
> >
> > Relationship with practical techniques: the main takeaway of our findings can be summarized as “if you can improve capability A, then capability B is also improved”. This statement does not distinguish between different fine-tuning strategies. Formally, if the weight of the edge $z_2 \to z_3$ has weight $w$, then it implies that we can get $w$ improvement *in average*, due to the presence of noise in our causal model. It is possible that, for instance, RLHF leads to more improvement than SFT. However, in this work, we try to capture general causal links between capabilities rather than concrete, quantitative dependencies on specific fine-tuning techniques.
> >
> > Targeting math without catastrophic forgetting: we feel that the description of that paragraph may cause some confusion. In the causal representation learning literature, there are two ways to change the value (or distribution) of a causal factor $z$. The first is through “single-node intervention”, meaning that only $z$ is changed; the remaining causal links remain unchanged. The second is through intervening on $z$’s causal parents.
> >
> > In our setting, these correspond to two different strategies if we want to improve a target task performance (say, math reasoning). The most natural way to instantiate the first approach is to fine-tune on math-related data. However, this typically results in performance decrease on other capabilities and is not ideal (actually, this violates the “single-node intervention” definition since other capability values are also changed). We briefly mentioned catastrophic forgetting as the main challenge here. The second strategy, which we explore in this paper, is to fine-tune on some other capability that causally affects math reasoning. Our analysis discovers instruction tuning as such capability and verifies it through SFT experiments.
> >
> > **Question #2: What is the justification for the specific definition of MIC in Definition 2? Could alternative CRL algorithms designed for multi-node interventions yield different results? How sensitive is the result to the specifics of the ICA implementation or the permutation alignment process in HCA (Algorithm 2)?**
> >
> > Definition of MIC: the introduction of MIC is motivated by the fact that on real-world data, the learned causal graph may not be exact. In Definition 1, we state the formal requirements for an exact causal graph. One most natural way to relax it is presented in Definition 2, where we allow for mild entanglement between the exogenous noise variables. Mathematically, this is characterized by how close the entangle matrix $U$ is close to identity. MIC allows us to quantify how well a specific dataset is fitted by a linear causal model, as hypothesized in our paper.
> >
> > Choices of CRL algorithm: our theoretical results (Appendix C) guarantees that with infinite samples the algorithm is exact. In the finite sample regime, this is not guaranteed. However, we can always measure the performance of the causal graph recovery via MIC. In our algorithm, we essentially exhaust all possibilities and pick one with the smallest error.
> >
> > Sensitivity to ICA and permutation alignment: as mentioned above, we exhaust all possible permutations to avoid missing any potential “good match”. This is feasible when $d=3$, but for larger $d$’s we probably need some heuristic method such as aligning greedily, and global optimality is not guaranteed. However, as we already emphasized, our causal analysis does not aim to give a final solution to the fine-tuning practice. It just reduces the “search space” and eventually should be verified by actual fine-tuning experiments.
> >
> > In our identifiability result, we assume that the ICA step is exact. We find it generally difficult to quantify the error induced by the ICA step,since the ground-truth is unknown. We use FastICA in Scikit-learn to complete this step, which is typically the default choice for ICA. Also, we intentionally select domains with relatively large sample sizes to avoid having large errors in this step.

---

> > > ### Author Response · Authors · 2025-11-14
> > >
> > > **Question #3: The paper links $z_1$  to pre-training compute (Fig 8). How exactly does pre-training compute act as a confounder influencing all capabilities, and how does HCA effectively control for this beyond simple regression adjustment?**
> > >
> > > Great question! The short answer is that we effectively control for these confounding factors via dividing the whole dataset into domains according to the base model.
> > >
> > > This is actually a crucial step in our analysis and its importance is elaborated in Section 2. An illustration of our setup could be found in Figure 1. Base model already contains information about pre-training compute, and because we have a separate causal graph for each domain, this confounding factor is essentially removed.
> > >
> > > Generally, one should expect that if the pretraining compute is larger, then all capabilities should tend to be stronger. For the specific factor $z_1$, note that it is modelled as a function of the base model (see Figure 1). Our Figure 8 shows that $z_1$ can actually be well-predicted by the pretraining compute only. This is interesting and reflects the popular “scaling law” types of study, since base model contains much more information other than the pretraining compute, such as the type of data used in pretraining.
> > >
> > > Our algorithm, HCA, controls for the pretraining compute via controlling for the base model, a stronger confounding factor.
> > >
> > > We thank the reviewer again for the effort in the reviewing process, and we look forward to hearing back from the reviewer in case there are any additional questions.

---

### Official Review · Reviewer_VK8r · 2025-11-01

**Soundness:** 3
**Presentation:** 3
**Contribution:** 3
**Rating:** 4
**Confidence:** 3

**Summary:**

The paper proposes Hierarchical Component Analysis (HCA), a causal representation learning framework to uncover hierarchical latent capabilities of LLMs using only observational benchmark data (specifically, the Open LLM Leaderboard). The key idea is to model benchmark performance as a linear transformation of latent capability factors, with a shared base model variable treated as a confounder.

**Strengths:**

1. This work provides a novel causal framing by introducing causal representation learning to interpret model capability hierarchies.

2. The method is based on well-motivated derivation of HCA with identifiable conditions and theoretical grounding.

3. Reliance on public leaderboard datasets ensures reproducibility and relevance for community benchmarking.

**Weaknesses:**

1. The hierarchy is inferred from only six benchmarks, so the results may not generalize.

2. No robustness study on the hyperparameters.

3. Table 2 is not referenced in the main text.

**Questions:**

1. The paper briefly mentions targeted fine-tuning, but the description in the main text lacks sufficient detail. Can the authors elaborate more or point me to related texts?

2. Is it possible to compare performance of the proposed method to that of other latent-factor discovery methods?

---

> ### Author Response · Authors · 2025-11-14
>
> We thank the reviewer for the insightful feedback. Below is our response to the weaknesses and questions.
>
> **Weakness #1: The hierarchy is inferred from only six benchmarks, so the results may not generalize.**
>
> We only consider six benchmarks because the Open LLM Leaderboard, the main data source that we rely on, only has evaluation data on six benchmarks. Nonetheless, in the Appendix we also present additional results based on the old Open LLM  Leaderboard and the MMLU by task leaderboard, that contain more benchmarks (albeit with different set of models being evaluated), proving that our method is widely applicable.
>
> **Weakness #2: No robustness study on the hyperparameters.**
>
> We conduct sensitivity analysis in Section H. We find that if we include more base model families into our study, the role of $z_1$ and $z_2$ can be different but the main causal relationship we discovered from “Instruction following” to “math reasoning” remains stable.
>
> **Weakness #3: Table 2 is not referenced in the main text.**
>
> Thanks for pointing this out! The description of this table starts at roughly line 424. The main message is that under instruction tuning, math reasoning performance tends to have non-negligible improvements. This is an important final step in the whole pipeline: verify the hypothesis via actual fine-tuning after forming it via causal analysis. We will update this paragraph with an explicit reference in a revised version.
>
> **Question #1: The paper briefly mentions targeted fine-tuning, but the description in the main text lacks sufficient detail. Can the authors elaborate more or point me to related texts?**
>
> By “targeted fine-tuning” we meant fine-tuning on a specific domain – math reasoning in our context. It appears in line 425 in the discussion about the latent factor $z_3$. On one hand, $z_3$ is highly correlated with mathematical reasoning, but efforts that directly target to optimize $z_3$ often hurt other capabilities. For instance, Qwen2.5-Math-7B achieves 24.6% accuracy on IFEval, much lower than the 33.7% accuracy of the Qwen2.5-7B base model. In the terminology of causal representation learning, this implies that a “single-node intervention” that improves $z_3$ while keeping $z_1, z_2$ invariant could be difficult to find. On the other hand, our work reveals a causal link between $z_2$ and $z_3$ so that improving $z_2$ is an alternative way to get improvement in $z_3$, as verified through SFT experiments.
>
> **Question #2: Is it possible to compare performance of the proposed method to that of other latent-factor discovery methods?**
>
> This is a great question. The short answer is no, because our problem formulation is unique and our method is carefully designed to match our formulation, rather than an arbitrary choice.
> What is unique in our formulation, compared with other factor modelling analysis (e.g. [1,2,3])? The main difference is that we model the capabilities as a structural causal model (SCM), where the ordering matters (e.g. causal link A -> B makes capabilities A and B asymmetric), while previous works do not take this approach and the order of capabilities don’t really matter.
>
> But why bother recovering such an SCM with a more complicated algorithm? The reason is that the precise motivation of this work is to discover *causal links between LLM capabilities*. Such causal links can be very useful to know for fine-tuning but are only accessible through controlled experiments in previous works, which can be expensive to conduct. By contrast, our work provides an alternative approach to discover such causal links purely based on observational data.
>
> And this is the main motivation behind our formulation. Other latent-factor discovery methods do not consider such a causal structure, so they certainly cannot be applied to our setting. That said, we introduced MIC and investigated principal component subspace distances (Figure 2(c)) to ensure that we can quantify the performance of our current method.
>
> [1]. Ruan, Yangjun, Chris J. Maddison, and Tatsunori B. Hashimoto. "Observational scaling laws and the predictability of langauge model performance." NeurIPS 2024.
>
> [2]. Ren, Richard, et al. "Safetywashing: Do AI Safety Benchmarks Actually Measure Safety Progress?." NeurIPS 2024.
>
> [3]. Polo, Felipe Maia, et al. "Sloth: scaling laws for LLM skills to predict multi-benchmark performance across families." NeurIPS 2025.
>
> We thank the reviewer again for the effort in the reviewing process, and we look forward to hearing back from the reviewer in case there are any additional questions.

---

### Meta-Review · Area_Chair_4okR · 2026-01-01

**Summary:**

This paper proposes a causal representation learning framework, namely Hierarchical Component Analysis, to reveal the hierarchical causal relationship among latent capabilities identified through pure observational data. By modeling the benchmark performance of several base model families as a linear transformation of latent capability factors, HCA uncovers a common SCM with the following dependency order: foundational general capability -> instruction following -> advanced math reasoning. This SCM provides valuable insights as how post-training on benchmarks could affect performance on other tasks.

Overall, this work offers an interesting and novel causal formalization of latent capabilities of LMs only via observational data. The proposed method and discoveries are supported by theoretical discussions as well as empirical validation. The uncovered hierarchical causal structure provides valuable guidance for post-training and LLM development.

Beyond those, the reviewers identified the following limitations:
- How the method could be applied to a richer set of datasets and tasks or LMs is not clear.
- It is not clear whether the method can generalize to more than three factors given the complexity of SCM computations. The scalability might become an issue.
- It is not clear how this method and findings can map to more realistic and advanced post-training strategies such as RLHF, other LM families, and architectures (such as MoE).

**Reviewer Concerns:**

Concerns being addressed:
- Robustness analysis and generalizability to other benchmarks and models are provided in appendix.
- Clarifications of MIC, choice/definition of latent factors
- Choice and definition of confounding factor
- Explanation of strong assumptions such as linearity of capability-performance mapping, the DAG structure of capabilities, and non-Gaussian noise for ICA
- Explanation of the relationship between HCA and prior algorithms

Outstanding concerns:
- Comparisons of HCA with other methods (but as pointed out by the authors, there is no suitable methods to be compared with)
- How the method could be applied to a richer set of datasets and tasks under the same LMs is not clear.
- It is not clear whether the method can generalize to more than three factors given the complexity of SCM computations. The scalability might become an issue.
- It is not clear how this method and findings can map to more realistic and advanced post-training strategies such as RLHF, other LM families, and architectures (such as MoE).
- Exclusion of models with different PC subspaces may limit the generalizability.

**Reviewer Scores:**

I might expect the review score being increased slightly if full participation is involved. I do acknowledge that the authors have made their effort in clarifying several points and answering the questions made by the reviewers. Certain concerns have been addressed, which may result in an increasing score (probably 4->5). The outstanding concerns as listed above are still remaining. Therefore, I suspect the final evaluation might be still mixing (negative and positive).

---

### Decision · Program_Chairs · 2026-01-26

Reject